# Language experience in LSF development: Behavioral evidence from a sentence repetition task

**Caroline Bogliotti**[1]*, **Hatice Aksen**[2], **Frédéric Isel**[1]

**1** Laboratoire MODYCO CNRS & Paris Nanterre University, Nanterre, France, **2** Laboratoire Structures Formelles du Langage CNRS & Saint Denis University, Paris, France

* caroline.bogliotti@parisnanterre.fr

**Data Availability Statement:** The anonymized data underlying the results presented in the study are available on the Nakala Platform (https://www.nakala.fr/data/11280/23b72fc9).

**Funding:** This research was partially supported by the Education, Audiovisual and Culture Executive

## Abstract

In psycholinguistics and clinical linguistics, the Sentence Repetition Task (SRT) is known to be a valuable tool to screen general language abilities in both spoken and signed languages. This task enables users to reliably and quickly assess linguistic abilities at different levels of linguistic analysis such as phonology, morphology, lexicon, and syntax. To evaluate sign language proficiency in deaf children using French Sign Language (LSF), we designed a new SRT comprising 20 LSF sentences. The task was administered to a cohort of 62 children– 34 native signers (6;09–12 years) and 28 non-native signers (6;08–12;08 years)–in order to study their general linguistic development as a function of age of sign language acquisition (AOA) and chronological age (CA). Previously, a group of 10 adult native signers was also evaluated with this task. As expected, our results showed a significant effect of AOA, indicating that the native signers repeated more signs and were more accurate than non-native signers. A similar pattern of results was found for CA. Furthermore, native signers made fewer phonological errors (i.e., handshape, movement, and location) than non-native signers. Finally, as shown in previous sign language studies, handshape and movement proved to be the most difficult parameters to master regardless of AOA and CA. Taken together, our findings support the assumption that AOA is a crucial factor in the development of phonological skills regardless of language modality (spoken vs. signed). This study thus constitutes a first step toward a theoretical description of the developmental trajectory in LSF, a hitherto understudied language.

## Introduction

Assessing language abilities in deaf people is a real challenge as their language mastery varies widely depending on factors such as their environment, health and socioeducational level. A direct consequence of this situation is that professionals such as teachers, clinicians, and researchers have to assess language skills almost case by case basis. The challenge is even greater since an assessment must satisfy the needs of each professional who works with Sign Languages (SL), not only measuring the individual's general proficiency in sign language but

Agency – EACEA (543264-LLP- 1-2013-1-ITKA2-KA2MP "SignMET" – Principal Investigator: Pasquale Rinaldi). Sign Language coders were paid by the EVASIGNE Project (Paris Lumière University Funding) (PI: Caroline Bogliotti & Marion Blondel). The funders had no role in study design, data collection and analysis, decision to publish, or preparation of the manuscript. No additional external funding was received for this study.

**Competing interests:** The authors have declared that no competing interests exist.

also providing a specific focus corresponding to the professional's needs (e.g., phonological development, proficiency level of second language learners of sign language, modeling sign language acquisition).

Since the 1970s, the Sentence Repetition Task (SRT) has frequently been used in spoken language assessment [1,2], and has proven efficient for measuring children's linguistic abilities in various populations (e.g., native speakers, second-language learners, bilinguals, children, adults with language disorders, socioeconomically disadvantaged individuals). The SRT is also known to be a reliable clinical marker of language impairment [3–10]. Concretely, the task involves repeating a sentence produced by a speaker (experimenter, teacher, etc.). The participant's repetition (or recall) should be as similar as possible to the sentence produced by the speaker. Once the sentence has been recalled, error types can be analyzed, thereby providing information about the level of language analysis that may disturbed. These errors are divided into phonological errors, omissions, substitutions, and deletions. As in spoken languages, errors in an SRT conducted in a sign language also constitute a reliable indicator of normal and delayed acquisition at linguistic levels of analysis (phonology, morphology, syntax, lexicon). Therefore, the SRT is viewed as a valuable screening tool for language disabilities. Moreover, it has several advantages: it is quick and easy to administer; it tests explicit, precisely specified linguistic structures; and it is not too time-consuming for professionals who have to assess and score in addition to teaching or providing therapy.

Although the SRT has proven to be a good tool for detecting delays or disorders in language development, no SRT study had previously been done on French Sign Language (LSF) development in deaf children. To fill this gap, we designed a behavioral experiment in LSF using this task with deaf children. Our goal was to study language development in LSF at different levels of analysis (phonological and morphosyntactic development, lexical knowledge) as a function of age of SL acquisition (AOA) and chronological age (CA) using an SRT. This study constitutes an innovative contribution to the sign language literature as it is the first attempt to investigate language development in LSF with a large cohort (*N* = 62) of deaf children. In addition to this developmental issue, our study introduces a new task to assess linguistic knowledge in children in an understudied language, namely LSF, the language used by deaf people in France. As in all sign languages that adapt to the specific visuo-gestural modality, LSF signers simultaneously use manual and non-manual articulators (hands, arms, chest, face, gaze) to produce sublexical and lexical signed units, and locate these signs in the signing space to set up the spatial grammar, that is, organize the syntactic relations between signs. The linguistic analysis of LSF is relatively young, and few grammars have been published (for a review paper, see [11]).

## The SRT in sign languages

To date, only a few studies have been carried out on sentence repetition ability in sign languages with deaf people [10,12–14]. Most such studies aimed to between distinguish deaf adults who had and had not acquired sign language as a native language and showed that the SRT constitutes a valuable tool to highlight error rates that vary according to signer group [13,15]. These inter-group distinctions are important because they show how the lack of early exposure to sign language can influence language development and the mastery of complex structures. Moreover, because the SRT involves both receptive and expressive skills, it is also known to be a good proxy for general linguistic abilities.

Mayberry and Fischer [16] ran the first SRT study with American Sign Language (ASL) adult signers, whose AOA varied. The authors observed that AOA had a strong impact on repetition skills, with the native signers recalling more sentences more accurately than the less

sign-exposed deaf adults. In addition, the types of errors differed according to AOA: while native signers produced more semantic errors, non-native signers produced more phonological errors, as they paid more attention to the surface structure (phonological shape) of the stimuli. Despite long experience (around 15 years) with sign language, non-native signers underperformed in recall accuracy and exhibited less accurate lexical and phonological skills, indicating the great importance of native exposure. Hauser et al. [15] used an SRT with an ASL sample to provide a test of ASL proficiency that would allow researchers to collect a full range of data from the heterogeneous community of deaf signers. The task comprised 40 ASL sentences varying in length and morphosyntactic complexity. Sentences were repeated by adolescents (12.5 to 14.1 years) and adults who were ASL natives or non-natives. The authors observed a strong effect of developmental age and of ASL AOA on general SRT scores. After this pilot study, they chose the 20 items that were most sensitive to native proficiency to build the final SRT version. This final task was used in Supalla et al.'s [14] study. Once again, AOA and length of exposure were found to influenced repetition performance. The most striking results concerned the types of errors and the strategies used when repeating. While fluent signers tended to preserve the semantics of the sentence, less fluent signers tended instead to repeat the surface structure. Cormier et al. [13] adapted the ASL-SRT to British Sign Language (BSL) and administered it to a group of 20 adults composed of 10 native signers, 5 early learners, and 5 non-native learners of BSL. The authors observed that the earlier the AOA, the more accurate the repetition skills. Marshall et al. [10] used the same task to assess repetition abilities in deaf children with Specific Language Impairment (SLI) in comparison to deaf controls who were non-native learners of BSL. They found that deaf children with SLI presented poorer repetition skills, were less accurate in sentence repetition (modifying the meaning or the order of signs), frequently omitted spatial morphology markers, and rarely used facial expressions. An interesting point was the pattern of errors, which was very similar to that observed with hearing children with SLI.

Recently, Rinaldi et al. [12] reported the results of an SRT with native and non-native deaf children and adults, who had all received intensive exposure to Italian Sign Language. They observed effects of both chronological development and AOA on repetition skills, and emphasized the difficulty the children had in acquiring and mastering the non-manual components.

## Does the SRT assess only language skills?

There is a lively debate concerning what abilities are needed to recall a sentence and what kind of capacities a repetition task assesses, and in particular whether memory or language processes are most involved in an SRT. Several researchers contend that memory skills underlie language skills and have demonstrated the role of working memory in language skills, in both neurotypical people and populations with disabilities [6,17–20]. Acheson and MacDonald [21,22] argued that linguistic knowledge and memory are intertwined, and that working memory cannot be separated from language comprehension and production. Along the same lines, Vargo and Black [23] ran an SRT with aphasic adults whose recall skills were well preserved. Their results showed that memory ability is involved but only slightly. The same results were obtained with children with SLI [24]. Devescovi and Caselli [25] investigated the relation between the SRT and working memory in preschool children and reported contradictory results; they argued that, even if the SRT partly relies on memory, strong linguistic abilities are necessary to recall sentences.

Several arguments have been proposed to show that SRT ability depends as much on language as on memory. The first concerns the evidence that all language levels are involved in the repetition process [26,27]. As soon as a speaker hears/sees the sentence to be repeated, he/

she builds a conceptual representation. This conceptual representation relies on several memory and linguistic processes to recall the sentence: sensorimotor processes (speech-sign perception and speech-sign production), phonological representations, lexical knowledge and grammatical encoding. Another argument in favor of the strong involvement of language skills concerns the complexity of the structures assessed: if the sentence reaches a certain length or structural complexity threshold, mere imitation is not sufficient, and robust linguistic representations are crucial [28]. In addition, some researchers have pointed out, repetition ability also depends on the speaker or signer's familiarity with the assessed language. Repetition of a two-sign sentence is easy for young signer children, non-fluent signers, and even non-signers, because repetition can be based on mimicking gestural components and not just on linguistic components. But repeating a syntactically complex sentence in sign language requires robust linguistic representations and good phonological and working memory abilities [2,10,21].

Natalicio [29] raised other critical issues regarding the naturalness of the repetition task. This task is often criticized for not being able to assess true language skills since repetition is ecologically irrelevant. By merely repeating a sentence, speakers do not experience a spontaneous interaction. To rehabilitate this task, Natalicio pointed out that most language-testing situations in a laboratory are not ecological, due to the presence of the experimenter, the recording equipment, the place of testing, the response and time constraints, and all the other variables that are absent in spontaneous interactions. Whatever the task, testing is not a natural event, but the linguistic behaviors identified by assessment still allow researchers to highlight a real language competence.

Another interesting point that we take into account here concerns the use of the gestural modality and the impact in terms of sensorimotor processing: cognitive abilities are very dependent on the gestural modality. Boutla et al. [30] showed that deaf adult signers have a shorter phonological memory span than hearing adults. This could be the consequence of the inherent processing abilities of SL: because they use a visual system, signers are likely more efficient at retaining spatial information than temporal information. However, it is temporality that is most frequently measured in phonological short-term memory trials, even when the assessed language is an SL. We can extend this observation to linguistic skills and we must be extremely careful not to confuse SL characteristics that may influence deaf signers' abilities with those abilities themselves. However, it should be noted that, as soon as linguistic structures or world knowledge become available, equivalent memory skills are expected irrespective of the language modality.

## Native sign language acquisition

In the current state of knowledge, it is assumed that typical sign language acquisition and development follow a similar trajectory to those of a spoken language [31–34]. When a child benefits from full exposure to an SL from birth, this is referred to as native acquisition. Several psycholinguistic studies have reported that signing children of deaf parents (i.e., native signers) show a linguistic development course similar to that of hearing children of hearing parents. For example, a lexical burst is observed around 16 to 20 months of age in SL development [35]. SL development starts at around 6 to 9 months with manual babbling, and production of the first signs is characterized by sign simplifications, substitutions, and reduplications depending on the children's motor limitations and the phonotactic constraints of their SL [36–38]. Like their speech-exposed counterparts, native signers produce their first signs around 10 to 12 months and also seem to organize their lexicon around semantic categories [35,39–41].

Regarding the acquisition of syntax, it is difficult to compare speech- and sign-exposed children because the SL-specific spatial syntax is typologically distinct from spoken languages. Mastery of spatial components in SL syntax that have no vocal counterparts makes it difficult

to compare acquisition in the two modalities [42,43]. Furthermore, some linguistic specificities related to the visuo-gestural modality have been reported in SL development, such as strong eye contact in infancy, manual babbling, absence of simultaneous production of gestures and signs (i.e., language with gesture), and a higher rate of production of predicates.

While the native acquisition of different SL is relatively well understood thanks to a rich empirical literature that is widely disseminated in the SL scientific community, to our knowledge, no study has been done on the developmental stages of LSF acquisition. Some research has been carried out on the acquisition of specific abilities such as pointing functions in an LSF signer deaf child [44], referential processes in children's narrations [45], LSF-French child bilingualism [46], gesture-sign development in a longitudinal study of four children [47], and morphosyntactic skills in children [37].

## Consequences of non-native sign language acquisition for language abilities

Native exposure to SL is far from the most common situation: 95% of deaf children are born to hearing parents who are not able to speak an SL with their child at birth and in the early years [48, for US statistics: 49], while others are born to deaf parents whose knowledge of sign language is incomplete or deficient [50–53]. This late exposure to sign language has consequences that are far from trivial, as language development is known to differ in native and non-native signers. While native signers, either deaf or hearing, with deaf parents grow up and acquire language in an efficient sign language environment, non-native signers usually encounter the gestural modality or sign language at school. Consequently, the linguistic environment has strong impacts on language acquisition and the linguistic skills observed in adulthood. As Cormier et al. [54] claimed, acquisition of a first language from birth will ensure native proficiency, whereas delayed first language acquisition prevents children from achieving complete acquisition. Therefore, most deaf children who learn SL experience language deprivation, the consequence of which is atypical language development.

We know now that lack of exposure to SL from birth may cause significant cognitive and linguistic delays [51,55–57]. Some studies have shown that deaf children exposed to SL late in childhood present unusual phonological development. For example, the first handshapes can be of different levels of complexity, and the production of these handshapes may depend on gestures that non-native learners produced before they were exposed to sign language [56,58]. In this study, it is specifically the impact of AOA on language development that we aim to examine. The same observation as for phonological development has been reported for the development of morphosyntactic structures [56,59,60]. In BSL sentences, Cormier et al. [54] used a grammaticality judgment task and found greater inaccuracy in non-native learners than in native ones; inaccuracy increased as a function of the age of the first exposure to BSL. Poor syntactic ability in sign language production was also reported by Ferjan-Ramirez et al. [51] in deaf teenagers with extremely late language acquisition (around 14 years), who therefore presented atypical sign language development, as indicated by a syntactic deficiency despite a large lexicon. The authors also noted that these teenagers learned signed vocabulary very quickly compared with younger native signers despite the same length of exposure. In contrast, their morphosyntactic skills were poor, with a childlike Mean Length of Utterance (MLU), around 2.08 signs for the most productive teenage signer aged 15;08 years [55]. This MLU is similar to that of a young child aged from 1;08 to 2;0 years [37].

In sum, most researchers report that early exposure to sign language provides benefits, even for second language (L2) literacy [60,61]. Early/native exposure should be distinguished from SL experience: even with 15 to 20 years of practice, non-native adults cannot behave like adults who are native signers [6,53].

## Assessing sign language and the development of signed abilities: A long road to success

It is necessary to provide benchmarks for the developmental trajectory of SL ability. To succeed in this, we must be able to assess the developmental trajectory. Unfortunately, as mentioned above, there is a lack of assessment tools for sign language development, and in particular for LSF. While several tests have been created to assess different linguistic levels and populations, it is not yet possible to perfectly describe the linguistic development of signing deaf people in its all aspects ([62] and http://www.signlang-assessment.info). Several attempts have been made to create tests in LSF but none of them are systematically used because of their lack of reliability. For example, the TELSF (Test for French Sign Language [63,64]) is considered to be too long to run and extremely difficult to score; furthermore, no successful adaptation of the LSF Receptive Skills Test has been developed [65]. Consequently, there is no commonly accepted test to measure LSF performance and proficiency. These tests did, however, enable us to highlight the constraints and difficulties encountered when adapting a language test [66]. Creating a tool that measures sign language skills is not an easy task. First, gestural languages require some technical adaptations to their physical and structural properties, and it is crucial to consider technical issues such as the presentation of visual stimuli or the recording of participant responses. In addition, to build an SRT, sign language linguists need to collaborate with deaf native signers to select suitable linguistic items and record them. We must also ensure that the assessment tool is easy to administer and score for all the professionals who will use it (i.e., speech therapists, researchers, teachers and specialist teachers, sign language teachers) and in all test situations (i.e., school, speech therapy, L2 sign language courses). Finally, we need to provide a reliable test that can rank a child on a clear, accurate language developmental scale [40].

## This study

This study examined language development at different levels of analysis (phonology, morphology, lexicon, syntax) in deaf children using French sign language, taking into account the AOA and CA, two determining factors in spoken language development. To assess sign language abilities in LSF-using children, we designed a first LSF Sentence Repetition Task, composed of 20 sentences. The SRT is known to assess linguistic abilities reliably and quickly. We administered our SRT to a cohort of 62 children comprising 34 native signers (6;09–12 years) and 28 non-native signers (6;08–12;08 years).

In this experimental study, in order to maintain some variety in our sentences, as in natural production, the children's ability to repeat signed sentences was assessed with 20 LSF sentences varying in linguistic complexity (i.e., length [2 to 6 signs], type of morphosyntactic units [e.g., absence/presence of classifiers, mouth gestures]). In accordance with previous sign language studies on AOA and SRT, non-native signers were expected to achieve a lower performance level than native signers on our SRT, and to make more lexical-semantic and phonological errors. In addition, older children were expected to perform better than younger ones. Finally, we expected that, at the same age, native signers would outperform non-native signers.

## Materials and method

### Participants

Sixty-two deaf children (38 girls) aged from 6;01 to 12;09 years old (mean age = 8;08 years, SD = 1;06 years) participated in the study. They had varied experience in LSF and different AOA, length of exposure to LSF, and quantity and quality of LSF input. (Table 1 shows the

**Table 1. Sociodemographic data on deaf children: 34 native signers and 28 non-native signers assessed with the SRT.**

| CA | AOA | Type of Hearing Aid | Length of exposure to LSF and / or French | Hearing Status of parents | Home Linguistic Environment |
|---|---|---|---|---|---|
| 06;02 | native | - | LSF family & LSF school | deaf—deaf | LSF |
| 06;04 | native | - | LSF family & LSF school | deaf—deaf | LSF |
| 06;05 | native | hearing device | LSF family & LSF school | deaf—hearing | LSF—Vocal French |
| 06;09 | native | - | LSF family & LSF school | deaf—deaf | LSF |
| 07;00 | native | - | LSF family & LSF school | deaf—deaf | LSF |
| 07;01 | native | - | LSF family & LSF school | deaf—deaf | LSF |
| 07;04 | native | - | LSF family & LSF school | deaf—deaf | LSF |
| 07;04 | native | - | LSF family & LSF school | deaf—deaf | LSF |
| 07;05 | native | - | LSF family & LSF school | deaf—deaf | LSF |
| 07;06 | native | - | LSF family & LSF school | deaf—deaf | LSF |
| 07;07 | native | - | LSF family & LSF school | deaf—deaf | LSF |
| 07;10 | native | CI | LSF—French family & LSF school | deaf—deaf | LSF |
| 08;01 | native | - | LSF family & LSF school | deaf—deaf | LSF |
| 08;03 | native | - | LSF family & LSF school | deaf—deaf | LSF |
| 08;03 | native | - | LSF family & LSF school | deaf—deaf | LSF |
| 08;08 | native | - | LSF family & LSF school | deaf—deaf | LSF |
| 08;08 | native | - | SL family & LSF school | deaf—deaf | Romanian SL * |
| 08;10 | native | - | LSF family & LSF school | deaf—deaf | LSF |
| 08;11 | native | - | LSF family & LSF school | deaf—deaf | LSF |
| 09;00 | native | - | LSF family & LSF school | deaf—deaf | LSF |
| 09;01 | native | - | LSF family & LSF school | deaf—deaf | LSF |
| 09;05 | native | - | LSF—GSL family & LSF school | deaf—deaf | Greek SL—LSF * |
| 09;10 | native | - | LSF family & LSF school | deaf—deaf | LSF |
| 10;00 | native | - | LSF family & LSF school | deaf—deaf | LSF |
| 10;02 | native | - | LSF family & LSF school | deaf—deaf | LSF |
| 10;03 | native | - | LSF family & LSF school | deaf—deaf | LSF |
| 10;03 | native | - | LSF family & LSF school | deaf—deaf | LSF |
| 10;04 | native | - | LSF family & LSF school | deaf—deaf | LSF |
| 10;06 | native | - | LSF family & LSF school | deaf—deaf | LSF |
| 10;07 | native | - | LSF family & LSF school | deaf—deaf | LSF |
| 10;07 | native | - | LSF family & LSF school | deaf—deaf | LSF |
| 10;07 | native | - | LSF family & LSF school | deaf—deaf | LSF |
| 11;02 | native | - | LSF family & LSF school | deaf—deaf | LSF |
| 12;00 | native | - | LSF family & LSF school | deaf—deaf | LSF |
| 06;01 | late | CI | kindergarten | hearing—hearing | French |
| 06;06 | late | CI | LSF non-dominant language | hearing—hearing | French |
| 06;09 | late | CI | LSF non-dominant language | hearing—hearing | French |
| 06;09 | late | CI | AOA 03;01—experience 03;08 | hearing—hearing | French |
| 06;10 | late | - | LSF non-dominant language | hearing—hearing | French |
| 07;00 | late | hearing device | AOA 03;04—experience 03;08 | hearing—hearing | French |
| 07;01 | late | CI | LSF dominant language | hearing—hearing | French |
| 07;02 | late | hearing device | *not known* | hearing—hearing | French & Berber |
| 07;03 | late | CI | not known | hearing—hearing | French |
| 07;05 | late | CI | LSF non-dominant language | hearing—hearing | French |
| 08;00 | late | 2 CI | AOA 04;06—experience 03;06 | hearing—hearing | French |
| 08;00 | late | - | kindergarden | hearing—hearing | French |
| 08;04 | late | hearing device | *not known* | hearing—hearing | French |

*(Continued)*

**Table 1.** (Continued)

| CA | AOA | Type of Hearing Aid | Length of exposure to LSF and / or French | Hearing Status of parents | Home Linguistic Environment |
|---|---|---|---|---|---|
| 08;11 | late | CI | *not known* | hearing—hearing | French |
| 09;01 | late | CI | *not known* | hearing—hearing | French |
| 09;07 | late | CI | *not known* | hearing—hearing | French |
| 09;08 | late | - | *not known* | hearing—hearing | French |
| 09;10 | late | hearing device | LSF non-dominant language | hearing—hearing | French |
| 09;11 | late | CI | *not known* | hearing—hearing | French |
| 09;11 | late | post linguistic deaf | elementary school | hearing—hearing | French & Swiss German |
| 10;01 | late | bi-CI | LSF non-dominant language | hearing—hearing | French |
| 10;05 | late | CI | *not known* | hearing—hearing | French |
| 10;08 | late | - | kindergarden | hearing—hearing | French |
| 10;08 | late | CI | elementary school | hearing—hearing | Kurdish & French L2 |
| 10;09 | late | - | *not known* | hearing—hearing | French |
| 11;00 | late | - | LSF non-dominant language | hearing—hearing | French |
| 11;01 | late | CI | *not known* | hearing—hearing | French |
| 12;09 | late | CI | *not known* | hearing—hearing | French |

The children's parents were asked to specify their hearing status (deaf or hearing), which language(s) they spoke or signed daily with their child, and what kind of exposure their child had to LSF and spoken French. This information allowed us to set up AOA groups.

* These children have LSF as the dominant language, as they have been educated in LSF-French bilingual school since the age of 3 years. In addition, deaf teachers assured us that these children are highly proficient in LSF. CA: chronological age; AOA: age of acquisition; CI: cochlear implant; LSF: French Sign Language.

biographical characteristics of the participants.) Thirty-four children were native signers (mean age = 8;75 years, SD = 1;06 years) who grew up in deaf families and had strong exposure to LSF from birth, while 28 children grew up in hearing families and were therefore considered non-native signers (mean age = 8;84 years, SD = 1;8 years). We conducted a *t*-test that failed to show a significant difference between the groups for CA ($t(60) = 0.16$, $p > .05$). Native and non-native signers received a bilingual education (written French and LSF), learned LSF at school, and used LSF as their preferred language. Note that all native children attended the same school, which ensured consistency with respect to the language environment/experience for all of them. According to the parental questionnaires and teacher reports, none of the children had other cognitive and/or social impairments. All children were given information about the goal of the study and informed that they could cease to participate in the experiment at any time if they wished. Before the experiment, their parents signed a consent form and were also informed about the purpose of the study and the exact nature of the experimental task. Before we ran the experiment with deaf children, 10 deaf adults who were all native signers of LSF were tested to ensure the validity of our SRT. The adults aged from 25 to 45 (mean age = 29;9 years; 3 males and 7 females) and grew up in different areas of France, ensuring real LSF expertise. Moreover, participants were matched for socioeconomic status. Before taking part in the study, the participants were individually informed about the experimental protocol and the data storage and anonymization procedure. They then gave their written consent. The data collected were anonymized by applying the European Data FAIR principle [67] in collaboration with HumaNum for the management of the experimental data. The study was approved by the local ethics committee of the Department of Psychology at Paris Nanterre University (SignMET. Project Number 543264-LLP-1-2013-1-IT-KA2-KA2MP) and was performed in accordance with the Declaration of Helsinki. The individuals appearing in the

pictures in this article gave written informed consent (as outlined in the PLOS consent form) to publish these case details.

## Sentence repetition task in LSF: Stimuli

The SRT in LSF consisted of 20 sentences, which were selected from a pilot study in which two deaf adults assessed the initial pool of 35 LSF sentences based on their naturalness. Four criteria were taken into consideration in satisfying naturalness: (1) grammaticality, (2) plausibility, (3) age-matched semantic content, and (4) saliency of signs for display on screen. To create our 35 LSF sentences, we initially relied on Rinaldi et al.'s [12] corpus. We adapted the Italian version of the SRT thanks to deaf native signers in order to create a culturally and linguistically well-designed pool of LSF sentences. The sentences varied in length and syntactic complexity. (Fig 1 and S1–S4 Videos shows some examples; the complete materials are presented in S1 Table). Given that length is not the sole marker of structural complexity in sign language (as in spoken language), several short sentences included complex morphosyntactic structures such as (1) classifiers (a category of signs with a non-specific meaning, expressed by a particular handshape that specifies—classifies—a referent with a particular property, such as two-legged, vehicles, etc. [68]), (2) dual predications, and (3) use of the non-dominant hand to maintain reference in the signing space. Note that long sentences may contain only simple morphosyntactic structures.

The lexical items chosen were those that any deaf child was likely to be familiar with. The sign rate was designed to be adapted to children's perception and recognition, and facial expressions were present but not strongly emphasized. All the sentences were checked in our team composed of deaf speakers and deaf and hearing linguists. One LSF speaker signed the instructions that were given prior to the experiment as well as the 20 sentences constituting the SRT. Post-experiment interviews revealed that the task was not considered too easy or too difficult.

## Procedure

The children were tested by deaf native signers or fluent hearing signers. Before starting the SRT, the experimenter engaged in a short interview with the children in order to make them feel comfortable for the following task. Testing was carried out in the school library. The SRT lasted only 10 minutes. The instructions were previously recorded and were presented on a laptop. The experimenter verified that the children had understood the task correctly, and if not, repeated the instructions a second time. The children were told to repeat verbatim each sentence of the LSF-SRT presented on the laptop. The children's repetitions were video-recorded. Children were instructed to repeat in front of the camera to ensure that their repetition was properly recorded. They could ask to see the sentence a second time if necessary and make self-corrections.

## Scoring

Fluent signers annotated all the data twice. All annotations were done in an Excel spreadsheet (Table 2). Because the task consisted in repeating exactly the same sentence as the one produced by the LSF speaker, we coded the presence or absence of linguistic variation between the model and the child's repetition.

The steps in our analysis were as follows: (1) Was a sign repeated or not? (2) If it was repeated, then was it different from the model? (3) If it was different from the model, then what kind of lexical variation was observed: (a) substitution_regionalism; (b) substitution_-other sign; (c) variant of the target sign? (4) If the repeated sign was a variant of the target sign

| Item | Sign Span | Syntactic Complexity | Sentence content and inflections GLOSS & *approximate translation* |
|---|---|---|---|
| 3 | 3 | Easy | FRIENDS – MEET – KISS *Friends meet and kiss each other* |
| 7 | 4 | Intermediate easy | BONE - SMALL (SASS) - DOG – DISAPPOINTED *The dog is disappointed because its bone is small* |
| 14 | 5 | Intermediate difficult | CHILDREN - HAT - CL: hat on the head – CL: put the hat on the child's head – CL: match the hat to the child's head *I take the hat that I have on my head, I put it on the child's head and I fit it to child's head.* |
| 16 | 5 | Complex | BOX – CANDY + CL: box – EAT + CL: box – ANY CANDY LEFT + CL: box - DISAPPOINTED + CL: box + CL: any candy left *I ate all the candies that were in the box and there's nothing left, so I am disappointed* |

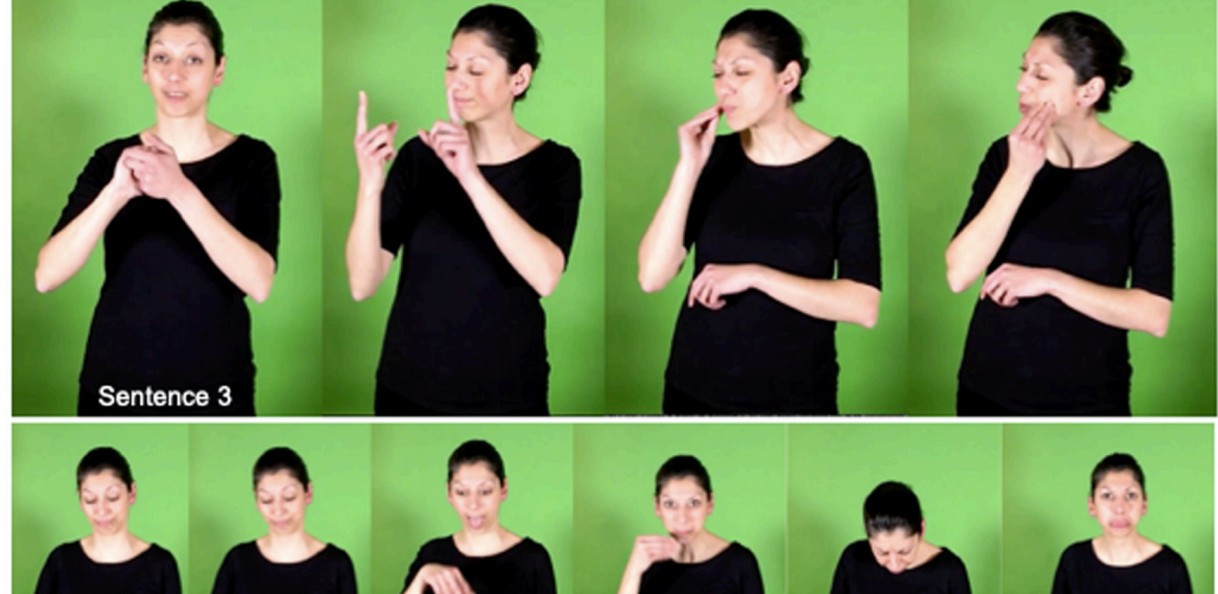

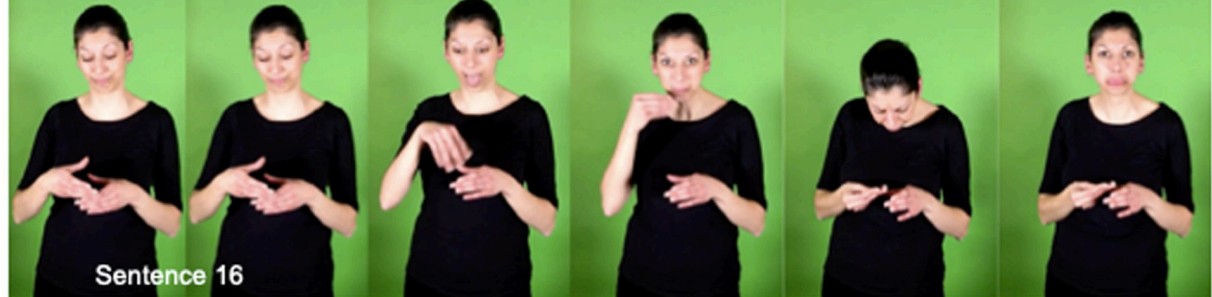

**Fig 1. Sentences with four levels of complexity.** Sentences of each level of syntactic complexity were presented in the Supporting Information (S1 Video: easy sentence; S2 Video: intermediate easy sentence; S3 Video: intermediate difficult sentence; S4 Video: complex sentence). All the sentences presented in the SRT are listed in the (S1 Table).

(called "phonological errors"), then what kind of manual parameter—that is, the phonological unit in LSF: handshape, movement, location, or orientation—was incorrectly reproduced?

As Table 2 shows, after viewing each sentence, the annotators reported for each sign in the sentence whether it was produced or not (YES or NO). If the sign was produced, coders reported whether the repetition of the sign was similar to or different from the model's (YES, the sign is different, or NO, the sign is similar). If the sign produced was different, the coders

**Table 2. Scoring grid for all signs.**

| GLOSS | Is the sign REPEATED? | If the sign is repeated, is it DIFFERENT from the model? | If the sign is DIFFERENT, which level is the difference located at? | | |
|---|---|---|---|---|---|
| CHILD (UL) | YES or NO | YES or NO | | Linguistic criteria | Difference |
| | | | Repeated sign | Substitution _ regionalism* | |
| | | | | Substitution _ another sign | |
| | | | | Variant of the target sign | |
| | | | UL—Manual Parameters | Handshape | |
| | | | | Movement | |
| | | | | Orientation | |
| | | | | Location | |
| | | | CL Dominant Hand | CL-DH size | |
| | | | | Handshape | |
| | | | | Movement | |
| | | | | Orientation | |
| | | | | CL-DH not held (reference not maintained) | |
| | | | | Inflection | |
| | | | | Wrong CL-DH location and structure maintained | |
| | | | | Wrong CL-DH location and structure not maintained | |
| | | | CL Non-Dominant Hand | CL-NDH size | |
| | | | | Handshape | |
| | | | | Movement | |
| | | | | Orientation | |
| | | | | CL-NDH not held (reference not maintained) | |
| | | | | Inflection | |
| | | | | Wrong CL-NDH location and structure maintained | |
| | | | | Wrong CL-NDH location and structure not maintained | |
| | | | Laterality | Dominant Hand—Non-Dominant Hand relation | |
| | | | Facial Expression | Lexical-semantic | |
| | | | | Grammatical + CL | |
| | | | Eye gaze | Eye gaze | |
| | | | Mouth actions | Mouthing | |
| | | | | Mouth gestures | |
| | | | Chest | Lexical-semantic | |
| | | | | Grammatical | |

CL: classifier; DH: dominant hand; NDH: non-dominant hand.

* Regionalisms represented a marginal proportion of repetitions. Non-manual parameters (laterality, facial expression, eye gaze, mouth actions and chest) have been coded but have not been taken in account for this paper. See S2 Fig to take notice of the target sign CHILD.

reported the linguistic level at which the sign varied. We decided to code each formal parameter that composed a sign. For this purpose, we separated manual parameters (lexical signs, classifiers produced with the dominant hand or non-dominant hand) from non-manual ones (facial expression, eye gaze, mouth actions, and chest posture).

Once the formal analyses were completed, we analyzed the repetition, taking into consideration two linguistic levels of analysis: phonological and lexico-semantic. First of all, we noted that the children clearly had difficulties repeating the non-manual parameters. We therefore decided to ignore non-manual variations as several factors in the experimental situation and the video-recording may have affected repetition performance: the children had to look at the experimenter whereas the sentence structure required them to direct the eye gaze to the hand, they were too young to spot fine actions such as mouthing, they were too young to master space perfectly, etc. At the end of scoring, for each child, our matrix displayed four columns, including (1) the number of signs produced, (2) the number of lexical errors among the produced signs, (3) the number of phonological errors, and (4) the types of phonological errors. Because sign language annotation is time-consuming, we employed two coders to score the children's repetition abilities. All the repetition material (62 recordings in all) was coded by two independent coders. Both coders were fluent in sign language and received the same annotation instructions with specific rules predetermined by the authors. The intercoder comparisons did not show reliable differences for any comparisons (Student $t$-test $p > .05$).

## Results

### Sign and sentence repetition

Two analyses of variance (ANOVAs) were run with AOA and CA as between-subject factors on the percentages of repeated signs (dependent variable 1, DV1) and of errors in sign repetition among the repeated signs (DV2). Moreover, the type of error (i.e., target sign, sign substitution and regionalism) was analyzed. The analyses run for each type of error were also conducted with AOA and CA as between-subject factors. Finally, additional analyses of repeated signs and errors among repeated signs were also run in deaf adults (control group). We first present the children's results followed by those of the adult control group. A descriptive comparison of children's and adults' performance is shown in Fig 2.

**Percentage of repeated signs.** The ANOVA revealed that the main effect of AOA was significant ($F(1,56) = 17.96$, $p < .001$, $\eta_p^2 = .240$; Fig 2). This indicates that, on average, the native signers repeated more signs (94.1%, SD = 5.9%) than the non-native ones (84.4%, SD = 13.7%). Moreover, the main effect of CA was also significant ($F(2,56) = 11.77$, $p < .001$, $\eta_p^2 = .296$). Planned comparisons revealed that on average the youngest children (6–7 years) repeated fewer signs (82.9%, SD = 15.2%) than the two older groups (for 8–9 and 10–12 years, respectively: 92.5%, SD = 6.33%, and 94.8%, SD = 4.56%; $F(1,56) = 23.23$, $p < .001$). In contrast, no significant difference was found between the two older groups ($F < 1$). The AOA × CA interaction was not significant ($F(2,56) = 1.96$, $p > .10$).

**Percentage of errors in repeated signs.** The ANOVA revealed a significant main effect of AOA ($F(1,56) = 29.18$, $p < .001$, $\eta_p^2 = .343$, Fig 2). This effect indicates that, on average, native signers were more accurate (34.5%, SD = 11.6%) when they repeated the LSF sentences than non-native signers (51.7%, SD = 15.2%). Furthermore, the main effect of CA also achieved significance ($F(2,56) = 5.89$, $p < .01$, $\eta_p^2 = .174$). Planned comparisons showed that the mean percentage of errors was significantly higher in the youngest group of children (49.7%, SD = 16.2%) in comparison with the two older groups of children together (8–9 years: 38.8%, SD = 12.9%; 10–12 years: 37.5%, SD = 15.5%; $F(1,56) = 11.79$, $p < .001$). No significant

**Age of Acquisition and Chronological Age**

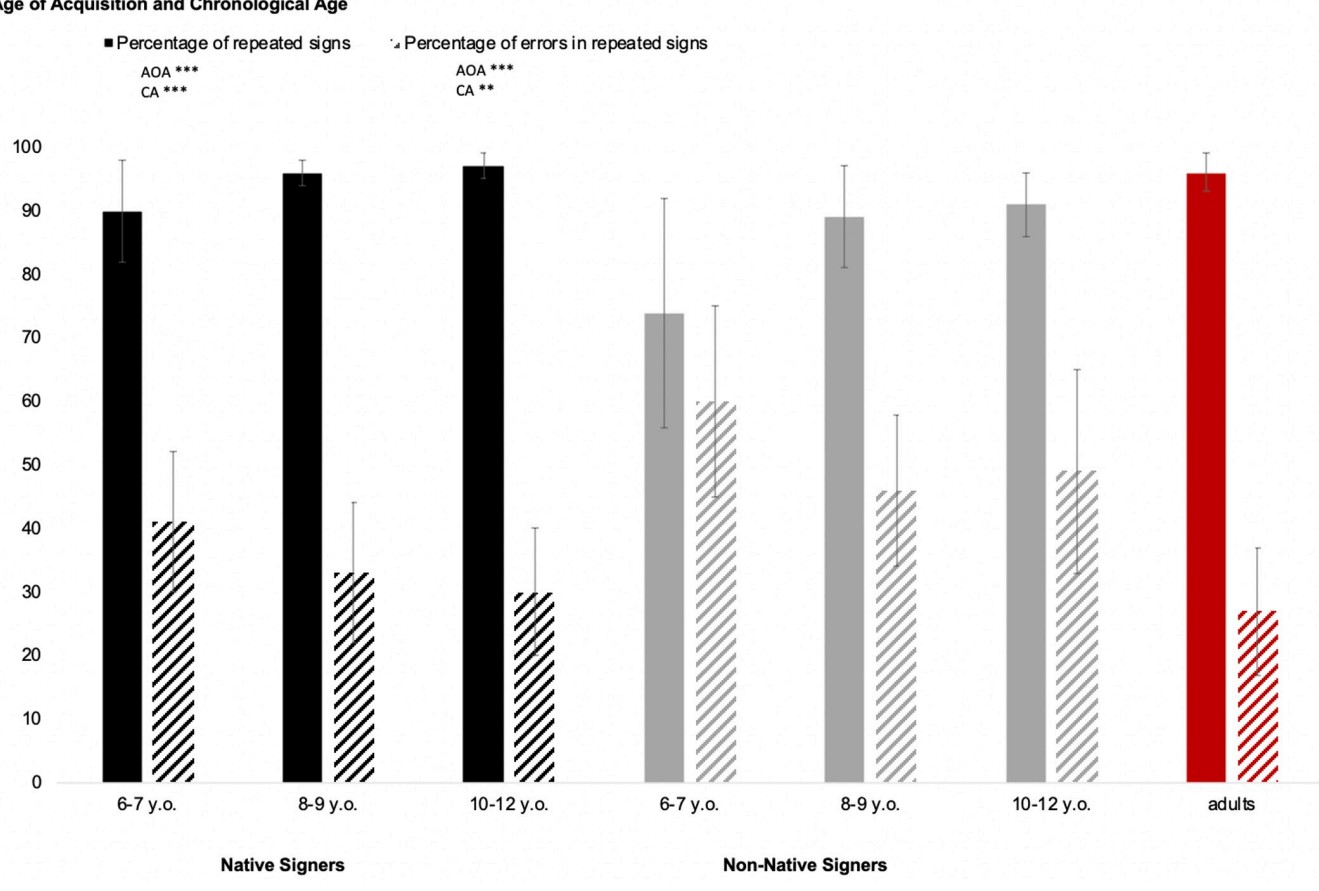

**Fig 2. Percentage of repeated signs and percentage of errors in repeated signs according to age of acquisition and chronological age.** Adults are considered as the control group (*** $p < .001$; ** $p < .01$; * $p < .05$; n.s. $p > .05$).

difference was found between the two older groups ($F < 1$). Finally, the interaction between AOA and CA was not significant ($F < 1$).

**Percentage of incorrect sign repetition according to sign category.** Incorrect sign repetition was classified into three types: (1) variant of the target sign (the repeated sign is easily recognizable as the one produced by the model but the child's repetition contains one or more errors); (2) substitution_other sign (i.e., another LSF sign has been substituted for the target sign); or (3) substitution_regionalism (i.e., the local sign has been substituted for the target sign). The percentage of type of errors was analyzed (Fig 3). The ANOVA showed that only the main effect of AOA was significant for regionalisms ($F(1,56) = 5.43$, $p < .05$, $\eta_p^2 = .80$), indicating that native signers used regionalisms more often to substitute for target signs (2.84%, SD = 2.92%) than non-native signers (1.28%, SD = 2.04%). No effect of CA ($F(2,56) = 1.79$, $p > .10$) or AOA × CA interaction ($F < 1$) was reported. Likewise, ANOVAs failed to reveal any significant effect for target signs (AOA: $F < 1$; CA: $F < 1$; AOA × CA interaction: $F < 1$) or for sign substitutions (AOA: $F(1,56) = 2.00$, $p > .10$; CA: $F < 1$; AOA × CA interaction: $F < 1$).

## Phonological errors in sign repetition

A 2 × 3 ANOVA with AOA (2 levels: native signers and non-native signers) and CA (3 levels: 6–7, 8–9 and 10–12 years) as between-subject factors was run on the number of phonological

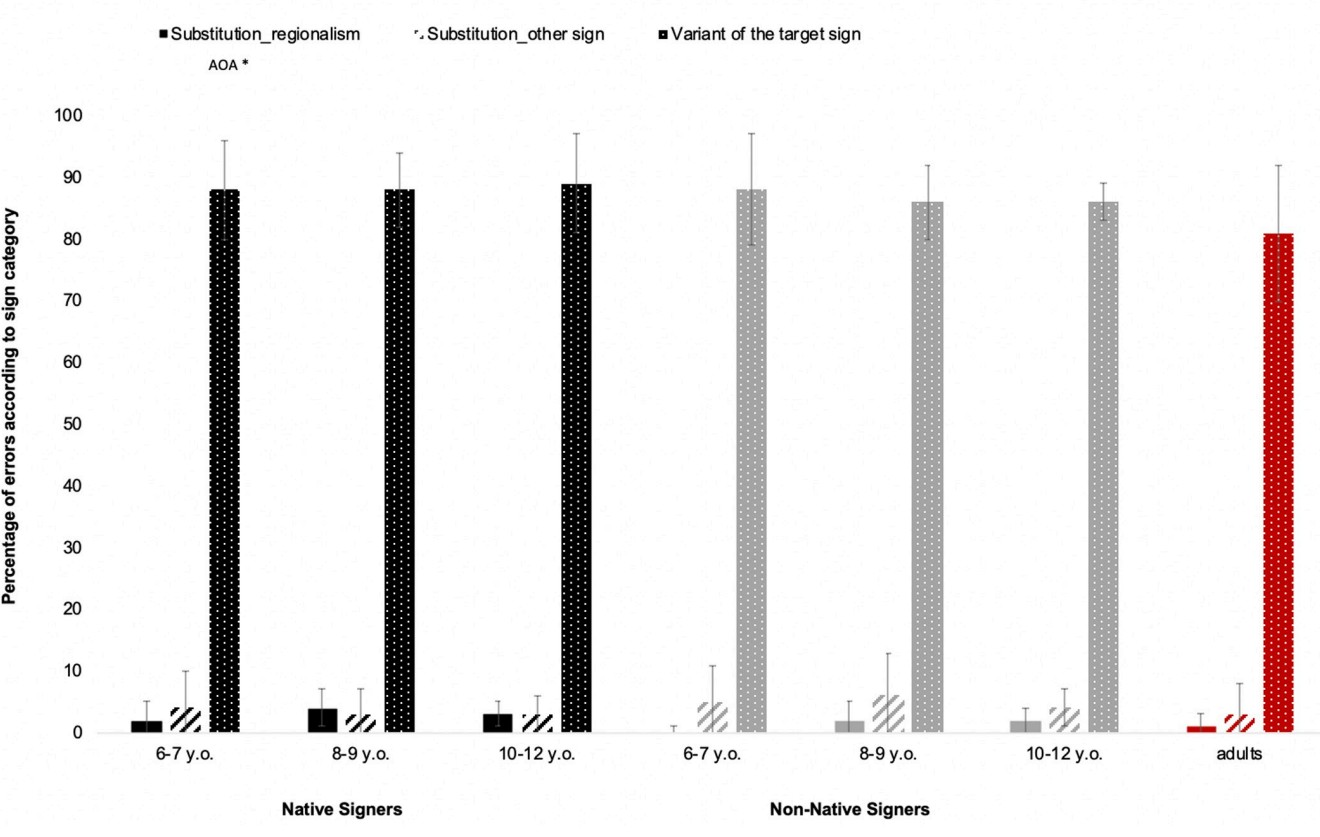

**Fig 3. Percentage of errors according to sign category (substitution_other sign, substitution_regionalism or variant of the target sign) with respect to age of acquisition and chronological age.** Adults are considered as the control group ($***$ $p < .001$; $**$ $p < .01$; $*$ $p < .05$; n.s. $p > .05$).

errors. Sign category (3 levels: Lexical Sign, Dominant-Hand Classifier and Non-Dominant-Hand Classifier) and Phonological parameters (3 levels: Handshape, Movement, Location) were considered as within-subject factors.

**Number of phonological errors.** The ANOVA showed that the main effect of AOA was significant ($F(1,56) = 37.03$, $p < .001$, $\eta_p^2 = .398$; Fig 4): made fewer phonological errors on average (M: 26 errors, SD = 12) than non-native signers (M: 50 errors, SD = 18). Neither the main effect of CA ($F < 1$) nor the AOA × CA interaction reached significance ($Fs < 1$).

**Number of phonological errors according to sign category.** Additional ANOVAs were conducted separately on the three different types of signs (Lexical Sign, Dominant-Hand Classifier and Non-Dominant-Hand Classifier). For Dominant-Hand Classifiers, a significant effect of AOA was found ($F(1,56) = 12.12$, $p < .001$, $\eta_p^2 = .178$; Fig 5), as well as a significant effect for the main factor CA ($F(2,56) = 4.71$, $p < .05$, $\eta_p^2 = .144$).

Planned comparisons showed that younger children made more phonological errors on average (28%, SD = 8.2%) on Dominant-Hand Classifiers than the two older groups ($F(1,56) = 9.65$, $p < .01$). In contrast, the two older groups made the same proportion of phonological errors on Dominant-Hand Classifiers (8–9 years: 23%, SD = 7.2%; 10–12 years: 20.6%, SD = 9.2%; $F < 1$). In addition, the AOA × CA interaction was not significant ($F(2,56) = 1.07$, $p > .10$).

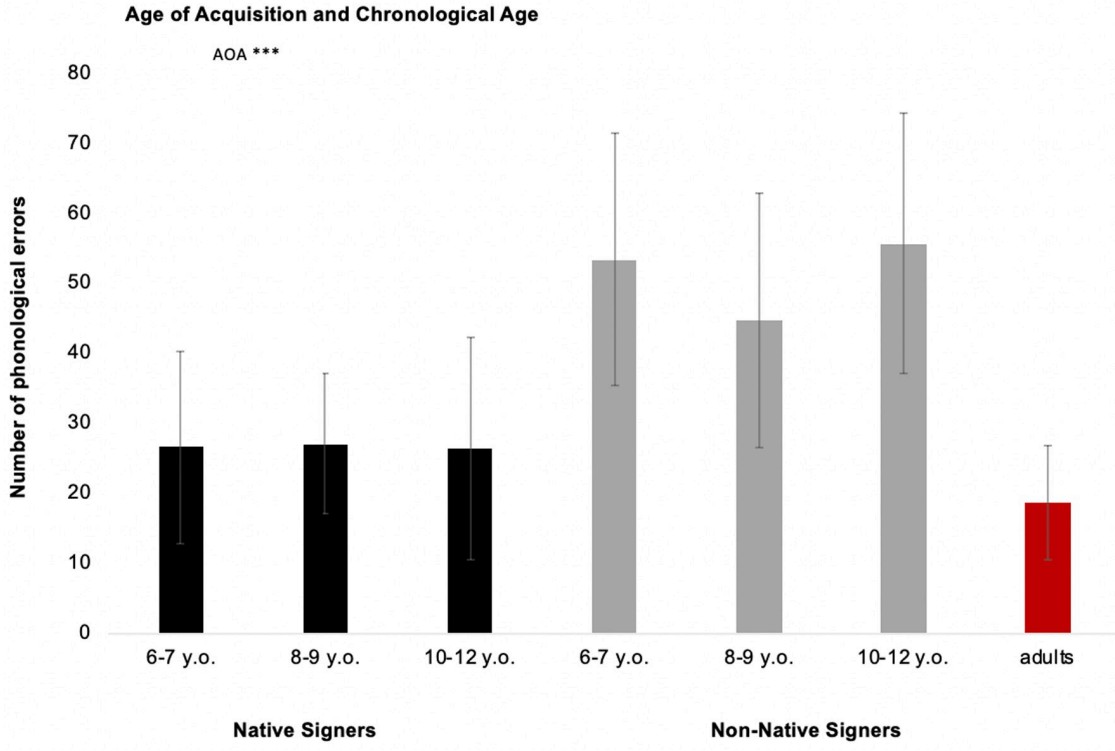

**Fig 4. Number of phonological errors with respect to age of acquisition and chronological age.** Adults are considered as the control group (*** $p < .001$; ** $p < .01$; * $p < .05$; n.s. $p > .05$).

For Lexical Signs, neither AOA ($F < 1$), CA ($F < 1$) nor the AOA × AC interaction was significant ($F(2,56) = 1.21$, $p > .10$). This indicates that all children made phonological errors involving Lexical Signs in the same proportion. The same conclusion applies to Non-Dominant-Hand Classifiers: no effect of AOA or CA and no AOA × CA interaction was observed (respectively, $F(1,56) = 2.28$, $p > .10$; $F < 1$; $F < 1$).

**Proportion of errors according to type of phonological parameter.**   The ANOVA revealed that the main effect of Parameter was significant, suggesting that children did not repeat manual parameters in the same way ($F(2,112) = 112.57$, $p < .001$, $\eta_p^2 = .06$; Fig 6). Planned comparisons showed that location was repeated more accurately than movement or handshape ($F(1,56) = 440.23$, $p < .001$; % of errors for location: 7.81%, SD = 7.33%; movement: 34.84%, SD = 11.96%; handshape: 35.58%, SD = 9.86%). In contrast, there was no difference in the proportion of errors for the movement and handshape parameters ($F(1,56) = 14.51$, $p > .10$). The main effect of AOA was not significant ($F(1,56) = 3.23$, $p > .05$), nor were the other effects (CA: $F < 1$; AOA × CA interaction: $F < 1$; Parameter × AOA interaction: $F(2,112) = 1.21$, $p > .05$; Parameter × CA interaction: $F < 1$; Parameter × CA × AOA interaction: $F < 1$).

## Adults' phonological abilities

The ANOVA showed a main effect of Sign category ($F(2,22) = 9.42$, $p < .001$, $\eta_p^2 = .460$), suggesting that adults made, on average, more phonological errors in Lexical Signs than in Classifiers (respectively 46%, SD = 10%; Dominant-Hand Classifier 26%, SD = 8%; Non-Dominant-Hand classifier: 27%, SD = 8%; Figs 2 to 6). A significant main effect was also found for

### Age of Acquisition and Chronological Age

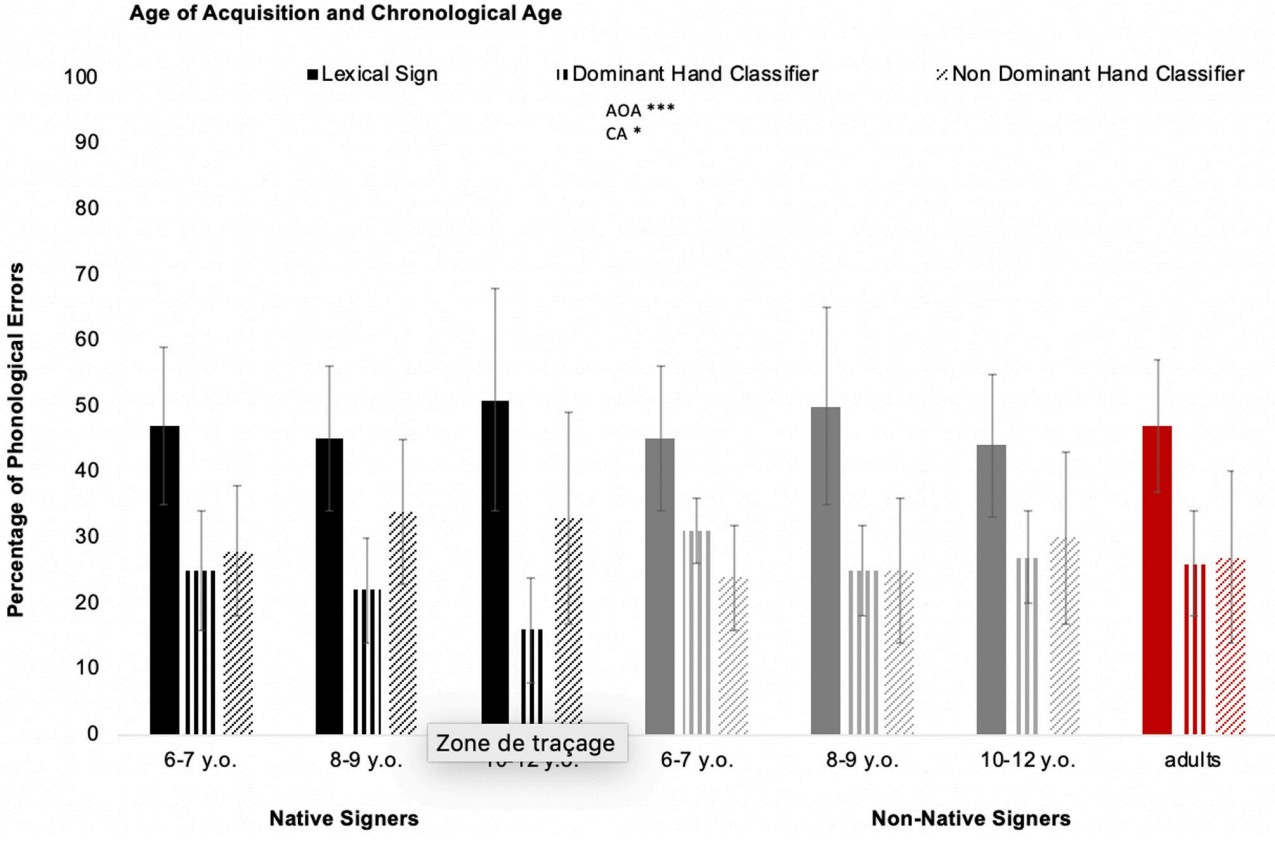

**Fig 5. Effect of age of acquisition and chronological age on phonological errors according to the sign category repeated: Lexical Sign, Dominant-Hand Classifier, and Non-Dominant-Hand Classifier.** Adults are considered as the control group (*** $p < .001$; ** $p < .01$; * $p < .05$; n.s. $p > .05$).

Phonological parameter ($F(2,22) = 14.28$, $p < .001$, $\eta_p^2 = .560$). Planned comparisons showed that location was repeated more accurately than movement and handshape (respectively, $F(1,1) = 11.86$, $p < .01$ and $F(1,11) = 23.07$, $p < .001$; % of errors for location: 5%, SD = 4%; movement: 27%, SD = 15%; handshape: 54%, SD = 20%). Contrary to children, there was a difference in the proportion of errors for the movement and handshape parameters ($F(1,11) = 7.89$, $p < .01$).

## Discussion

This study aimed to assess LSF abilities in children to trace their developmental trajectory. For this purpose, different levels of analysis (phonological and morphosyntactic development, lexical knowledge) were investigated as a function of AOA and CA using an SRT. The choice to use an SRT was motivated by the fact that this task constitutes a reliable indicator of normal or delayed acquisition that can examine phonological, morphological or lexical processing. Therefore, the SRT is viewed as a valuable screening tool for language disabilities. Our work is the first contribution to investigate the development of LSF in a large cohort of deaf children. In addition to this developmental issue, the strength of our study was that it introduced a new task to assess sign language knowledge in children who use an understudied language, namely LSF. To do this this, 34 native signers (6;09–12 years) and 28 non-native signers (6;08–12;08 years) in addition to 10 deaf adult controls were tested. As usual in developmental studies, we

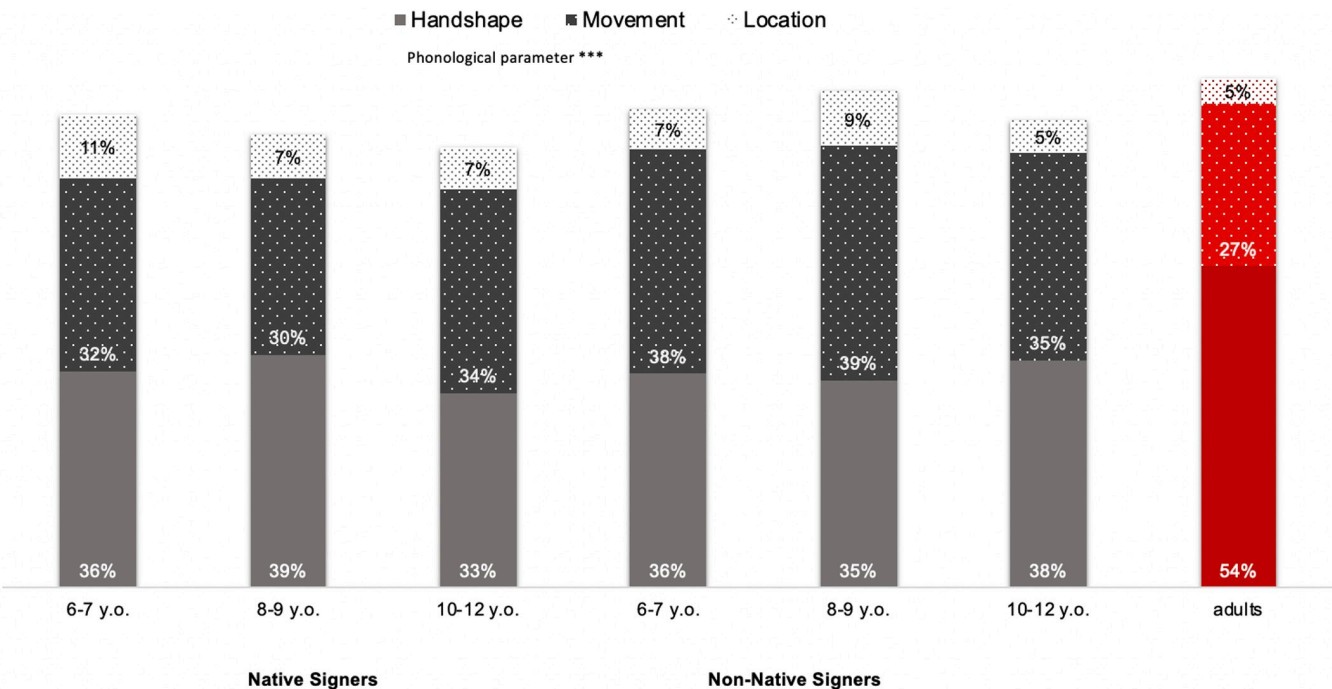

**Fig 6. Proportion of phonological errors made deaf signing children according to age of acquisition, chronological age and type of manual parameter.** Adults are considered as the control group (*** $p < .001$; ** $p < .01$; * $p < .05$; n.s. $p > .05$).

wanted to replicate the CA effect for LSF. The systematic variation of AOA allowed us to better understand the extent to which extend sign language development is affected by late acquisition. Children's abilities to repeat signed sentences were assessed with 20 LSF sentences that were carefully designed to vary in linguistic complexity (i.e., length [2 to 6 signs], types of morphosyntactic units [e.g., classifiers, mouth gestures]).

Regarding general repetition abilities, we were able to show that native signers generally succeeded better in repeating the sentences than non-native signers. This is attested by native signers' significantly better repetition abilities in comparison with non-native ones. Furthermore, with respect to CA, we found that, on average, younger children (6–7 years) made more errors in the SRT than the older ones (8–9 and 10–12 years). Surprisingly, the two older groups of signers performed equally well. One possible interpretation of this latter result is that at age 8 or 9 children already have good repetition abilities. An additional argument is that adults presented similar SRT error rates to the older children (see Fig 3).

At the lexical level, neither AOA nor CA had significant effects. The only effect found was that native signers produced more regionalisms than non-native signers did. One possible explanation of this effect could be the geographic area where we tested native signers. We went to the south of France, where native signers tend to produce regionalisms. As in Rinaldi et al.'s study [12], very few substitutions were found in our study. This result may be the consequence of the immediacy of the SRT recall task. Unlike a delayed repetition task [9], the immediate recall task does not involve basing the repetition on semantic grounds but rather on a "surface" mechanism such as mimicry, without integrating semantic content. If the task had been a delayed recall task, with a delay between the perception and repetition of the sentences, signers would have needed a semantic representation of the sentence in order to repeat it. In addition,

delayed repetition involves a higher working memory load to keep the information active. This might have resulted in the repetition of a correct sentence favoring the semantic content, rather than the formal characteristics produced by the model.

At the morphosyntactic level, like Rinaldi et al. [12], we found no differences in the effects of AOA and CA on morphosyntactic abilities. The SRT does not seem to be an appropriate task to assess the development of the syntactic and morphosyntactic aspects of language. It measures these aspects globally, but not accurately. Like Polišenská et al. [27], we assume that immediate SRTs test lexical phonology and morphosyntax more than semantic conceptual understanding. Although an SRT cannot perfectly test morphosyntactic aspects, it can capture the influences of CA and AOA, and thus it can be considered a screening task to measure over-all SL ability. A usage-based explanation can be suggested for this finding: structures with which speakers/signers have a lot of experience may be repeated better [69], explaining the bet-ter skills of native signers and older children. Several factors are known to affect the age of acquisition of the signs, vocabulary size and overall lexical development in the deaf population, such as the parents' hearing status, the age of identification of deafness, and the age of first exposure with sign language [70].

At the phonological level, unlike Rinaldi et al. [12], we decided to focus on manual parame-ters since our recordings did not allow us to do a fine-grained analysis of the non-manual com-ponents. Several children looked at the experimenter more than the camera, with the unfavorable consequence that we were unable to assess facial expression correctly; other chil-dren were intimidated by the assessment situation and did not produce typical facial expres-sions, although they did produce them during the interview.

We are aware that linguists may find it questionable to invoke phonology in the domain of signed languages. However, since the first structural studies of ASL [71,72], sign language lin-guists consider phonology to constitute not just the organization of speech sounds but rather the organization of minimal units in all languages, even gestural ones. Some attempts have already been made to describe sign languages phonologically [73–76] and claimed that signs can be segmented into meaningful sublexical elements. These sublexical elements, currently called parameters (i.e., handshape, movement, location and orientation), can be considered as phonological units. Parameter substitutions in a given sign can be compared to phoneme sub-stitutions in a word, and switching from one parameter to another makes it possible to oppose minimal pairs in all sign languages: for example, the sign CAKE, described as an "S hand" for *handshape*, with a "double short tapping movement" on the "cheek" for *movement* and *loca-tion*, can be opposed to the sign SOAP if one substitutes an "oscillating" movement for the "double short tapping movement" (see S1 Fig). An important issue in sign language studies concerns the role of modality in sign language acquisition, and more specifically whether the phonological development is sign-specific or not. Some researchers suggest that phonological processing is multimodal—or amodal—and involves representations that go beyond the sen-sory modalities [50,77,78]. Several sign language studies suggest that this language modality involves similar neural systems to those that support spoken language. While we do not claim that speech and sign processing are identical, this research indicates that sign language produc-tion and comprehension seem to rely on a similar left-lateralized neural network, whatever the level of processing involved, phonological, semantic, or syntactic [79–83].

Our results showed a significant effect of AOA on phonological skills, as indicated by a lower rate of phonological errors (i.e., handshape, movement, and location) in native signers than in non-native signers. However, both native and non-native signers exhibited the same pattern of results (handshape = movement > location). Interestingly, we also found that, regardless of AOA and CA, location was the best mastered phonological parameter. In con-trast, the pattern of results observed in adults was slightly different from that in children

(handshape > movement > location). In sum, analysis of the adults' phonological skills confirmed that the task was sensitive enough to capture phonological differences between the three parameters.

To summarize, our data highlight that the SRT in LSF appears to be a reliable task for detecting differences in phonological abilities as a function of AOA and CA.

As with spoken languages, early signed-phonological development depends on ease of production but also on the frequency of occurrence of both sign and parameter. While several studies have focused on early sign language development [84–88], no benchmarks concerning phonological development in older children are available in the literature. The *handshape* parameter—the shape of the hand when producing the sign—is the most difficult one to acquire. In a sign, different handshapes involve different numbers of features, which determine the markedness of particular handshapes. Marked handshapes are phonologically more complex (i.e., contain the greatest number of selected features) and motorically more difficult than unmarked ones. Unmarked handshapes involve the fewest selected features. Four stages in handshape development have been observed in ASL [89]. These stages have also been reported in another sign language [BSL: 90, 91], but no study has yet investigated this issue in LSF. Several studies have provided evidence that frequent handshapes in early signs are among the easiest to produce [88], and that unmarked handshapes are acquired and mastered early [85,89,92,93]. Unmarked handshapes are also often made with the non-dominant hand because they are the simplest to articulate. These unmarked handshapes are preserved in sign language aphasia [91]. Marked handshapes are more phonologically complex and are acquired later.

The *movement* parameter can be classified in two categories: path movement, which involves movement of the hand and the arm, and internal movement, which involves movement of the wrist and fingers. The initiation, type and temporal aspects of the movements performed by these four body parts are phonological: oscillating, waving, arc, straight line, curved, repeated, accelerated, etc. [94,95]. Because of its dynamic aspects and the more or less fine-grained mobilization of different body parts, the mastery of movement is driven by general motor skills (for a review, see [34]). Finally, *location* is the place in the signing space where the signer produces the sign: head and face, torso, shoulder, arm and neutral space. Because of its strong perceptual saliency and ease of articulation, this parameter tends to be the most accurately reproduced in the early stages of development [34,43,84–87,96].

Consistent with the previous literature, we observed that movement and handshape are the most complex phonological parameters to acquire and that location is mastered early [84–86,96]. Like Mann et al. [88], we showed that handshape and movement are still not accurately mastered by older children, whether native or non-native signers. As other researchers have already suggested, these repetition errors can be attributed to the motor aspects of these parameters, which are undoubtedly more complex [64,82]. In addition, the large inventory of handshapes and movements, in contrast to the location parameter, may influence the frequency of occurrence of each one, causing poorer mastery. In contrast, location is acquired faster because it does not require a fine-grained motor demand, and the anatomical understanding and cognitive representation of the body mean that it can be mastered more easily [96]. Furthermore, Caselli and Cohen-Goldberg [97] reported that location is a more robust parameter owing to its perceptual saliency and higher sublexical frequency.

Regarding the repetition of classifiers, native signers made fewer errors on classifiers executed with the dominant hand. They repeated them better, probably because they use this type of unit more frequently. For non-dominant-hand classifiers, there were too few tokens in our material to reach a conclusion about the influence of AOA or CA on correct repetition. We observed that only dominant-hand classifiers were affected by AOA and, to a lesser extent,

CA: native signers and older children repeated these units better. We suggest that this result is a consequence of frequency of use. Native and older signers produce dominant-hand classifiers more frequently—evidence of their morphosyntactic development and sign language mastery. Consequently, when they have to repeat them in an SRT, the phonological pattern of these units is more accurate.

Another cue that allowed us to observe the poorer quality of sign language among non-native signers concerns the sign stream. In non-native signers, the sign rate and pace were slow and non-fluent, with many incomplete movements. These characteristics were not taken into account in the scoring; the coders were asked to give a score from 0 to 5 in order to judge the quality of repetition, regardless of the signers' lexical and phonological skills. Coders did not know whether the participants were native signers or not. As expected, non-native signers were rated much lower, with a mean score of 2.9/5, whereas native signers had a mean score of 3.5/5. The intercoder comparisons did not show reliable differences for qualitative judgments (Student $t$-test $p > .05$).

To conclude, we were pleased to see that our adaptation of the SRT for LSF proved successful. The task highlighted differences in repetition abilities between native and non-native signers of LSF. With time, our study will provide a new LSF screening tool for the clinical, educative and scientific communities. We must ensure that the assessment tool is easy to administer and score for all the professionals who will use it (i.e., speech therapists, researchers, teachers and specialist teachers, sign language teachers) in all test situations (i.e., school, speech therapy, L2 sign language courses). Last, we need to provide a reliable test that can rank children on a clear and reliable language developmental scale [66].

## Future work

Our results provide partial support for the validity of the LSF-SRT, as reflected by the different significant effects of CA and AOA. However, we will not formulate firm conclusions about the validity issue until additional data from further studies are at hand. Beyond the standardization and calibration of the new SRT screening tool in LSF, the first version of which was presented in this article, further work should add qualitative analyses. The first analysis could be a phonetic analysis, as a measure of the phonetic complexity of unsuccessfully copied signs. Another analysis could involve measuring the sign stream (number of signs per minute), given that non-native signers seem to sign more slowly. In addition, to improve the quality of assessment, we could increase the complexity of sentences and the types of morphosyntactic structures used in the test. The SRT must be complemented with more accurate assessments to measure language ability or diagnose a language disorder in sign language.

We do not yet have an accurate representation of what kinds of linguistic abilities are mastered at a given age, or which levels can be expected to be achieved successfully in language acquisition and instruction. To the best of our knowledge, there are no reliable benchmarks of the developmental stages of language acquisition in deaf children who use LSF. This study is a first attempt to answer this question, but we need additional tools to evaluate deaf children's efficiency level in different linguistic domains. In future, we should be able to provide tools that make it possible to plot language development curves, especially during crucial periods of language acquisition, in order to quickly detect disorders. Another fruitful perspective would consist in testing children with dysfunctional acquisition such as non-native signers and children with SLI, for whom sign language data are rare and very heterogeneous. We have to determine whether sign repetition abilities may predict other language skills. Further investigations will combine these repetition data with narrative production based on a cartoon (Caët

& Blondel, in prep.) to assess to what extent production skills can be predicted from repetition abilities.

## Supporting information

**S1 Video. Easy sentence.**
(TXT)

**S2 Video. Intermediate easy sentence.**
(TXT)

**S3 Video. Intermediate difficult sentence.**
(TXT)

**S4 Video. Complex sentence.**
(TXT)

**S1 Table. All sentences.** SASS. Size and Shape Specifiers.
(DOCX)

**S1 Fig.** Minimal pairs in LSF in which changing the internal movement (a. CAKE/SOAP), the handshape (b. GIRL/NICE) or the location (c. IDEA/YELLOW) changes the meaning.
(TIFF)

**S2 Fig. Target sign /CHILD/. Coders had to compare this sign to the sign produced by participants.**
(TIF)

## Acknowledgments

First, we wish to thank all the deaf people who collaborated with us in order to produce well-formed LSF sentences and tested the children with us: Hatice Aksen, Celine Fortuna, Saliha Heouaine, Yves Prud'homme, and Clémentine Caron. We would also like to thank Aliyah Morgenstern for coordinating the French part of the SignMet project, as well as the researchers and PhD students who helped us to collect the data: Laetitia Puissant-Schontz, Anaïs Lisette, Anna Safar, Marion Blondel, and Stephanie Caët. Thanks to Vanessa Andrieu and Marie-Paule Kerrelhas, deaf teachers in Ramonville Saint-Agne, the teachers and speech therapists of the CRA (Hearing Rehabilitation Centre, Rouen) and CELEM (Language Education Centre for Hard of Hearing Children, Paris). Thanks also to the coders who helped us with the time-consuming work of data scoring. Finally, thanks to Philomene Perin for the pictures of minimal pairs.

## Author Contributions

**Conceptualization:** Caroline Bogliotti.

**Funding acquisition:** Caroline Bogliotti.

**Investigation:** Caroline Bogliotti.

**Methodology:** Caroline Bogliotti, Hatice Aksen.

**Supervision:** Caroline Bogliotti.

**Writing – original draft:** Caroline Bogliotti, Frédéric Isel.

**Writing – review & editing:** Caroline Bogliotti, Frédéric Isel.

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
