## [Decision Letter · Decision Letter 0]

27 Apr 2020

PONE-D-20-04317

Language experience in LSF development: Behavioural evidence from a sentence repetition task

PLOS ONE

Dear Mrs Bogliotti,

Thank you for submitting your manuscript to PLOS ONE. After careful consideration, we feel that it has merit but does not fully meet PLOS ONE’s publication criteria as it currently stands. Therefore, we invite you to submit a revised version of the manuscript that addresses the points raised during the review process.

Many of the points raised are very straightforward, such as ensuring that the English is proofread by a native speaker and that stylistic conventions (such as the use of abbreviations, and using decimal points rather than commas (for example, 7.45, not 7,45) are obeyed. The remaining points mostly ask for additional information and explanation about the methods, together with recommendations for some additional statistical tests. I am sure that none of these changes will pose any problems and look forward to receiving your revised version. We would appreciate receiving your revised manuscript by Jun 11 2020 11:59PM. To enhance the reproducibility of your results, we recommend that if applicable you deposit your laboratory protocols in protocols.io, where a protocol can be assigned its own identifier (DOI) such that it can be cited independently in the future. For instructions see: http://journals.plos.org/plosone/s/submission-guidelines#loc-laboratory-protocols

We look forward to receiving your revised manuscript.

Kind regards,

Bencie Woll, PhD

Academic Editor

PLOS ONE

Additional Editor Comments:

The development of an SRT for French Sign Language is an imprtant step both for researchers of LSF and for practitioners seeking a suitable assessment tool for use in educational settings where children are acquiring LSF. The authors have made considerable changes in content and focus from the original drafts and this has led to a much-improved paper, with very positive reviews. Both reviewers, however, ask for a number of changes and additions of information before the paper can be finally accepted. All of these are very straightforward. I look forward to receiving the final revised version and to its publication.

2. Thank you for stating the following beneath the Acknowledgments Section of your manuscript:

'Funding

This research was partially supported by the Education, Audiovisual and Culture Executive Agency – EACEA (543264-LLP- 1-2013-1-ITKA2-KA2MP “SignMET” – Principal Investigator: Pasquale Rinaldi). Sign Language coders were paid by the EVASIGNE Project (Paris Lumière University Funding) (PI: Caroline Bogliotti & Marion Blondel)'

'The funders had no role in study design, data collection and analysis, decision to publish, or preparation of the manuscript'

3. Your ethics statement must appear in the Methods section of your manuscript. If your ethics statement is written in any section besides the Methods, please move it to the Methods section and delete it from any other section. Please also ensure that your ethics statement is included in your manuscript, as the ethics section of your online submission will not be published alongside your manuscript.

4. We note that Figures 2 and 7 and Table 2 include an image of a person. 

Reviewers' comments:

Reviewer's Responses to Questions

**Comments to the Author**

1. Is the manuscript technically sound, and do the data support the conclusions?

Reviewer #1: Partly

Reviewer #2: Yes

2. Has the statistical analysis been performed appropriately and rigorously? 

Reviewer #1: Yes

Reviewer #2: Yes

3. Have the authors made all data underlying the findings in their manuscript fully available?

Reviewer #1: Yes

Reviewer #2: Yes

4. Is the manuscript presented in an intelligible fashion and written in standard English?

Reviewer #1: Yes

Reviewer #2: No

5. Review Comments to the Author

Reviewer #1: The project team developed a sentence repetition task (SRT) in French Sign Language (LSF) and tested the psychometric soundness of the LSF-SRT. The authors found that children who are native signers made significantly more repetitions than non-native signers and that younger children made more errors than older children. These finds are consistent with the findings of other language fluency tests.

The introduction was well written with an appropriate literature review. The authors adequately and thoroughly described past research on the use of the SRT in signed languages and the cognitive processes that the test measures including how these tests are more sensitive to language fluency than working memory skills.

While the age ranges of the native and late signers appear similar, please compute and share the results of a t-test to show that the two groups did not differ in their chronological ages.

The actual AOA of the subjects were not listed in Table 1 except “native” or “late. It would be better to include the actual AOA. Table 1 seems to suggest that at least one of the “native” signers has deaf parents that sign Romanian SL at home. It is unclear why this child was included in this study as a native LSF signer. And, another subject has one parent that uses Greek Sign Language and the other uses SLF. Similarly, three of the children in the “non-native” group are from spoken language bilingual families. At the present, we do not know if the sign language acquisition trajectory is the same for children who grow up in a home that does not use the same spoken/written language used at school compared to those who grow up in a family that uses the same spoken/written language used at school. If all of these children were removed from the study, are the results the same?

How was the initial pool of 35 LSF sentences created? What were the qualifications of the two deaf adults that repeated the initial pool of 35 LSF sentences and provided judgements? Were they native signers? Do they have psychology or linguistic backgrounds? What was taken into consideration in the selection of the “best” or most natural 20 sentences?

The reviewer is happy that the authors included Table 2 to give readers an idea of how repetitions were evaluated. This is helpful for other teams who wish to develop an SRT in their local sign language.

A description of the coders’ backgrounds would be helpful if another team wants to replicate the study. Were the coders deaf and native SLF signers? Since each subjects’ repetitions were coded by two coders, please provide the inter-rater reliability between coders. This is an important piece of the test’s psychometric soundness and the authors have the data available.

The reviewer got confused when reading the percentage of repeated signs then the percentage of errors in repeated signs. The percentage of repeated signs analyses included signs that were repeated incorrectly but the reviewer was expecting the first analysis to be on correct repetitions. The authors did not appear to analyze the AOA and CA effects on correct repetitions without errors. This analysis needs to be conducted.

When the number of phonological errors was reported, was that the total sample number of errors (in case of native signers, 26 errors, SD = 12) or the mean number of errors per native signing subject? Please provide the mean but it was not clear if it was the mean or sum.

It was noted that the native signers used regional variations in their repetitions. It is difficult to develop an SRT with no possible regional variations. If the observed regional variations were the same among a number of native signers then the team might want to consider it an appropriate/correct response rather than an error. This makes the reviewer wonder about the test instructions—were the test takers told to make exact repetitions and given an example of a variation and told why it is wrong and necessary to give an exact repetition? Some other sign language SRTs do this to minimize variations in repetitions. Please describe your test instructions.

Since chronological age (CA) is a continuous variable but was transformed to a grouping variable for the ANOVA results. It might be worthwhile to include the correlation between CA and percent correct repetitions so the results could be compared to other SRT studies.

The operational definition of “frozen signs” needs to be provided and the significance of analyzing dominant and non-dominant hand classifiers. Since this is being analyzed, it would be appropriate to report how many of these subjects were right-hand dominant.

Adults were included in the study and were called “control.” But, they did not really serve as a control group. If anything, the native signers were the control group for the late signers. The data from the adult group could be kept or omitted for a different manuscript as it did not really help the authors document the psychometric soundness of the LSF-SRT for children unless the authors wanted to compare the results of young native signers with adult native signers.

The use of “late” for non-native signers has been criticized in the past because “late” sounds more like a judgement compared to “non-native,” please consider changing the terms.

Similar to the introduction, the discussion was thorough and helpful.

The authors suggest that the LSF-SRT is a tool that clinicians and educators could use in the future. But, the authors did not thoroughly describe who their coders were. If their coders were deaf native signers with linguistics knowledge, then it might not be possible for hearing professionals or non-native signers to code results similar to the coders in this study.

Editorial feedback: Line 111, SLI was mentioned but it should be spelled out the first time used for the naïve reader. And, 132, “SLI children” should be changed to “children with SLI” (also, see line 112). Specific Language Impairment was spelled out (without its abbreviation) in the final section, Future Work. Please consider revising.

Claims made the sentence that starts with line 190 should be cited.

The reviewer did not review earlier submissions of the manuscript but read the authors’ responses to the past reviewers’ comments. It appears that the authors addressed most of the past reviewers’ concerns. The manuscript does appear to be more focused on test development and validity than phonology. The authors do not talk about validity at all although their results provide partial support on the validity of the LSF-SRT that they developed.

Reviewer #2: The manuscript has the potential to offer a valuable addition to research into LSF, and the use of a sentence repetition task as an assessment tool. It contributes to and confirms previous findings, whilst helping to bring LSF study in-line with studies in other sign languages. The notable number of native LSF participants from deaf families included in the study is also commendable.

However, I believe there are several issues that need to be addressed by the authors before it would be suitable for publication.

General comments:

• I propose that the article be proofread and corrected by a native English speaker. Whilst the English is mostly clear and understandable, it is not currently at a suitable standard for publishing in PLOS ONE, with many errors and omitted words, so substantial editing is needed.

• Care needs to be taken with consistency in using abbreviations. e.g. LSF, SL and SLI are frequently used in both long and short form. e.g. line 180-181 “SL development” is immediately followed by “sign language development”

• Sections of the manuscript appear to be repetitious. Eg. the start of the discussion section, whilst intended as a recap for the reader, is very similar to the abstract.

Introduction

• Line 111 is the first mention of ‘SLI’ so please correct to “specific language impairment (SLI)” for clarity.

• Line 88 - Likewise, explain ‘ASL’ as ‘American Sign Language’ in the first instance for the reader, unless it can be assumed the readers will be familiar with the abbreviation.

Methodology

• Little information is given as to how the sentences for the SRT were created. Did the authors adapt sentences from other SRTs, or did they create new sentences? A pilot study is mentioned (line 313) from which 20 sentences were selected for their “naturalness”. I am unclear as to what is meant by this and think it requires some clarification. Given that the focus of the manuscript is the utilisation of an SRT, more information about the creation of the sentences would be useful. The description provided of the linguistic content of sentences, along with complexity level is good.

• Participants - the inclusion of 10 deaf native signing adults is a useful addition to the study for comparison, however limited information is given about them other than their age range. As they are being used as a comparative group, please provide more detail on their ages (and mean age), and sex etc.

• Procedure - line 337 states testing took place in the school library, suggesting that all of the participants attended the same school. If this is the case, it may be a relevant point to make for the study as it offers some control to the language environment/experience of all of the children whilst they are at school. It may be a strength to the study to make this clear, if it is the case.

• If the children do all attend the same school, Table 1 on page 14 detailing children’s linguistic environment should clarify that this refers to their “home language environment”.

• Line 380-381 First states there are 6 coders, then states there are 2 coders - I assume the 2 independent coders were used for inter-rater reliability, but this is slightly unclear, please rephrase and add the reliability rating if available.

Results

• The asterisks indicating significance levels appear to be missing from the figures.

• Lines 431 and 432 contain commas instead of decimal points (2,84% instead of 2.84%). Proofreading needed.

Discussion

• Lines 671-674 appear to state that there are no known developmental stages of language acquisition in deaf children and that the current study is a first attempt to address this issue. This statement is not true. I assume that the authors intend this statement to relate to LSF, but it is unclear to the reader so needs to be clarified or rewritten.

• If lines 671-674 are, in fact, in relation to LSF, I am not convinced that this study addresses the question of developmental stages of language acquisition in LSF, beyond the finding that native or early LSF acquisition results in higher accuracy on the task.

• Line 650 - the participants’ fluency was rated by coders on a scale of 0-5. Were the coders blind to which group the participants were in (native or late learners), and was any inter-rater reliability undertaken? Only the average score for early signers is provided in the text, please provide the rest of the data for this measurement.

• Lines 539-540 regarding regionalisms - were participants all tested in the same region, or from different areas of France? if participants were all from the same area (or school) would the authors predict differing regionalisms in their signing?

6. PLOS authors have the option to publish the peer review history of their article (what does this mean?). If published, this will include your full peer review and any attached files.

Reviewer #1: Yes: pH

Reviewer #2: No

---

## [Author Response · Author response to Decision Letter 0]

29 May 2020

PONE-D-20-04317

Language experience in LSF development: Behavioural evidence from a sentence repetition task

PLOS ONE

Rebuttal Letter 

Answers to the Editor request 

Comment Editor 1#. Please ensure that your manuscript meets PLOS ONE's style requirements, including those for file naming. The PLOS ONE style templates can be found at

Response Editor 1# We checked templates once again.

Comment Editor 2# Thank you for stating the following beneath the Acknowledgments Section of your manuscript:

'Funding

This research was partially supported by the Education, Audiovisual and Culture Executive Agency – EACEA (543264-LLP- 1-2013-1-ITKA2-KA2MP “SignMET” – Principal Investigator: Pasquale Rinaldi). Sign Language coders were paid by the EVASIGNE Project (Paris Lumière University Funding) (PI: Caroline Bogliotti & Marion Blondel)'

'The funders had no role in study design, data collection and analysis, decision to publish, or preparation of the manuscript'

a. Please clarify the sources of funding (financial or material support) for your study. List the grants or organizations that supported your study, including funding received from your institution.

d. If you did not receive any funding for this study, please state: “The authors received no specific funding for this work.”

Response E2 # Thank you for this point. We now suppress all information related to the funding in the manuscript. We will put them directly in the Funding Statement section of the online submission form. > XXX

Comment E3#. Your ethics statement must appear in the Methods section of your manuscript. If your ethics statement is written in any section besides the Methods, please move it to the Methods section and delete it from any other section. Please also ensure that your ethics statement is included in your manuscript, as the ethics section of your online submission will not be published alongside your manuscript.

Response E2# We did this change.

Comment E3 # 4. We note that Figures 2 and 7 and Table 2 include an image of a person. 

Response E3# The individuals appearing in the figure 2 is Hatice Aken, coauthor of the paper, and she had already signed the Consent For. She’s the model in the SRT. The individual appearing in figure 7 is Philomène Perin and provided a Consent Form too. These two forms were already uploaded on the PLOS platform. Moreover, we added, as you suggested, this information in the Method section.

Answers to the Reviewer 1 request 

Reviewer #1: The project team developed a sentence repetition task (SRT) in French Sign Language (LSF) and tested the psychometric soundness of the LSF-SRT. The authors found that children who are native signers made significantly more repetitions than non-native signers and that younger children made more errors than older children. These finds are consistent with the findings of other language fluency tests.

The introduction was well written with an appropriate literature review. The authors adequately and thoroughly described past research on the use of the SRT in signed languages and the cognitive processes that the test measures including how these tests are more sensitive to language fluency than working memory skills.

Comment 1# While the age ranges of the native and late signers appear similar, please compute and share the results of a t-test to show that the two groups did not differ in their chronological ages.

Response 1# Thank you for this remark. We conducted a t-test which shows no difference between the groups for CA (t(60) = 0.16, p > .05).

Comment 2# The actual AOA of the subjects were not listed in Table 1 except “native” or “late”. It would be better to include the actual AOA. 

Response 2# We understand it would have been interesting to provide this information. However, for native, it is not necessary because they have LSF from birth. In addition, hearing parents of late signers have difficulties to estimate the real onset of exposure due to different learning context. Consequently, this information was not available for us.

Comment 3# Table 1 seems to suggest that at least one of the “native” signers has deaf parents that sign Romanian SL at home. It is unclear why this child was included in this study as a native LSF signer. And, another subject has one parent that uses Greek Sign Language and the other uses SLF. 

Response 3 # Thank you for pointing out this fact. We were aware about these two cases. However, we decided to include them in our sample for the following reasons. These children have LSF as dominant language, as they are schooled in LSF – French bilingual school since the age of 3 years. In addition, deaf teachers ensured us that these children are highly proficient in LSF. 

Furthermore, one of them has a native LSF input (French Sign Language mother with Greek Sign Language father). We add commentaries in the legend of the table in order to disambiguate this issue.

Comment 4 # Similarly, three of the children in the “non-native” group are from spoken language bilingual families. At the present, we do not know if the sign language acquisition trajectory is the same for children who grow up in a home that does not use the same spoken/written language used at school compared to those who grow up in a family that uses the same spoken/written language used at school. If all of these children were removed from the study, are the results the same?

Response 4 # Thank you for this commentary. However, the three mentioned children, due to their native deafness, did not access to spoken languages of their parents. Consequently, we reasonably assumed that they had a similar sign language acquisition trajectory than the other children with no bilingualism at home. 

Comment 5# How was the initial pool of 35 LSF sentences created?

Response 5 # To create our 35 LSF sentences, we initially relied on the corpus of Rinaldi et al. (2018). We adapted the Italian version of the SRT thanks to deaf native signers in order to create a well-designed LSF culturally and linguistically pool of sentences. We added this information in the Stimuli section of the revised version : “The SRT in LSF consisted of 20 sentences, which were selected from a pilot study in which two deaf adults assessed the initial pool of 35 LSF sentences based on their naturalness” (line 329) 

Rinaldi, P., Caselli, M. C., Lucioli, T., Lamano, L., & Volterra, V. (2018). Sign Language Skills Assessed Through a Sentence Reproduction Task. The Journal of Deaf Studies and Deaf Education, 23(4), 408 421. 

Comment 6# What were the qualifications of the two deaf adults that repeated the initial pool of 35 LSF sentences and provided judgements? Were they native signers? Do they have psychology or linguistic backgrounds?

Response 6# The two signers involved in the above-mentioned pilot study were native signers. They were recruited in our sign language linguistics department or work as assistant-engineer in a research lab.

Comment 7# What was taken into consideration in the selection of the “best” or most natural 20 sentences?

Response 7# Four criteria were taken into consideration for satisfying naturalness: 1) grammaticality, 2) plausibility, 3) age-matched semantic content, 4) saliency of signs for display on screen. We added this sentence in the revised version line 332.

Comment 8#The reviewer is happy that the authors included Table 2 to give readers an idea of how repetitions were evaluated. This is helpful for other teams who wish to develop an SRT in their local sign language.

Response 8# Thank you very much for this encouraging comment. The coding grid has been several times modified which occasioned a huge work.

Comment 9# A description of the coders’ backgrounds would be helpful if another team wants to replicate the study. Were the coders deaf and native SLF signers?

Response 9# We thanks the reviewer for this suggestion. Coders were hearing fluent signers who study in sign language linguistics department. When coders need information about a sign or a regionalism, we ask to deaf signers of the team to help them to disambiguate signs difficult to perceive. These latter cases were rare.

Comment 10# Since each subjects’ repetitions were coded by two coders, please provide the inter-rater reliability between coders. This is an important piece of the test’s psychometric soundness and the authors have the data available.

Response 10# we agree this is a relevant information to add. 

Analysis with Student t-test shows that they were no significant cross-coder differences for the following comparisons: 

- % of repeated signs (coder 1: 89.5% vs. coder 2: 90%), t(122) = 0.2, p > .05

- % of errors in repeated signs (coder 1: 41.8% vs. coder 2: 42.6%) t(122) = 0.24 p > .05

- Number of phonological errors (coder 1: 23.3% vs. coder 2: 25.6%), t(122) = 0.56, p > .0

- % errors of handshape (coder 1: 11.6% vs. coder 2: 12.2%), t(122) = 0.66, p > .05

- % errors of movement (coder 1: 11.1 % vs. coder 2: 12.1%), t(122) = 1.11, p > .05

- % errors of location (coder 1: 8.8%. vs. coder 2: 6.7%), t(122)=1.10, p > .05

Comment 11# The reviewer got confused when reading the percentage of repeated signs then the percentage of errors in repeated signs. The percentage of repeated signs analyses included signs that were repeated incorrectly but the reviewer was expecting the first analysis to be on correct repetitions. The authors did not appear to analyze the AOA and CA effects on correct repetitions without errors. This analysis needs to be conducted.

Response 11# Sorry for this confusing and we will try now to explain our methodological choice. For our study, what is critical is to target which kind of phonological errors (handshape, movement, location) will occur most frequently in deaf children. For this purpose, we decided to focus on percentage of repeated sign presenting errors. Among these wrong repeated signs, we have classified in the three type of phonological errors.

Comment 12 # When the number of phonological errors was reported, was that the total sample number of errors (in case of native signers, 26 errors, SD = 12) or the mean number of errors per native signing subject? Please provide the mean but it was not clear if it was the mean or sum.

Response 12# Statistically, the first analysis consisted to compute for each participant the sum of phonological errors. The second analysis lead us to calculate the ratio of each phonological error types (handshape, movement, location).

Comment 13# It was noted that the native signers used regional variations in their repetitions. It is difficult to develop an SRT with no possible regional variations. If the observed regional variations were the same among a number of native signers then the team might want to consider it an appropriate/correct response rather than an error. 

Response 13# Thank you for this remark. We separated regionalisms from the correct responses. However, we did not judge these regionalisms as errors as they were found in different repetitions. 

To explain our rationale to the reviewer: as our scoring grid was built by taking into account the criterion of an ‘exact repetition’, then only when the target sign was produced with a phonological variation in comparison to the model, it was considered as an error (i.e. variant of the target sign; Figure 3). Consequently, regionalism did not satisfy the “criterion error”.

Comment 14# This makes the reviewer wonder about the test instructions—were the test takers told to make exact repetitions and given an example of a variation and told why it is wrong and necessary to give an exact repetition? Some other sign language SRTs do this to minimize variations in repetitions. Please describe your test instructions.

Response 14# In our study, instructions given to the participants were clearly formulated: the participant have to look the sentence and must to wait the signer has finished to repeat as precisely as possible according the model. One hypothesis for explaining the emergence of regionalism is the frequency of some regionalism that frequency could probably not inhibit despite the instruction.

Comment 15# Since chronological age (CA) is a continuous variable but was transformed to a grouping variable for the ANOVA results. It might be worthwhile to include the correlation between CA and percent correct repetitions so the results could be compared to other SRT studies.

Response 15 # We ran the correlation between CA and percent of repeated signs with error. A correlation rp= -0.2 was found indicating that higher the CA, the lower the percentage of errors. However, this correlation was only marginally significant (.05 < p <.10)

Comment 16# The operational definition of “frozen signs” needs to be provided

Response 16# We agree and to avoid misunderstanding of the reader, we replace frozen by lexical. Indeed, it is well established now that SL have two types of lexicon: the core native lexicon, in which are classified the frozen sign, also called lexical signs, that refer to the permanent lexicon in sign languages. This lexicon includes signs that are highly stable, standardized in form and meaning, and frequently used in the sign language. By contrast, the non-native core lexicon in which we found item as depicting signs (classifiers, role shift) that are not stabilized in the sign language, highly variable and weakly lexicalized. The non-core native signs may move into the core lexicon over time (reference).

Comment 17# and the significance of analyzing dominant and non-dominant hand classifiers. Since this is being analyzed, it would be appropriate to report how many of these subjects were right-hand dominant.

Response 17# the rationale to separate Dominant and Non-Dominant Hand classifier was to study at which level of complexity the children were able to repeat. Note that the dominant hand classifiers are more frequent than the non-dominant hand classifier. Moreover, in general classifiers are units emerging during the LSF language development. Most of participants were right-hand dominant.

Comment 18# Adults were included in the study and were called “control.” But, they did not really serve as a control group. If anything, the native signers were the control group for the late signers. The data from the adult group could be kept or omitted for a different manuscript as it did not really help the authors document the psychometric soundness of the LSF-SRT for children unless the authors wanted to compare the results of young native signers with adult native signers.

Response 18 # We agree with reviewer that native signers could be the control group. That is the rationale we followed in the previous version of the manuscript. However, we were asked by a reviewer to add this control group. We found this suggestion fruitful as a reference group for this task. Moreover, it is usual in clinical linguistics to add such adult data in order to assess the validity of test.

Comment 19# The use of “late” for non-native signers has been criticized in the past because “late” sounds more like a judgement compared to “non-native,” please consider changing the terms. 

Response 19# We totally agree and we thank the reviewer. We now in the revised version, we replace ‘late’ by ‘non-native’ signers in opposition to native.

Similar to the introduction, the discussion was thorough and helpful.

Comment 20# The authors suggest that the LSF-SRT is a tool that clinicians and educators could use in the future. But the authors did not thoroughly describe who their coders were. If their coders were deaf native signers with linguistics knowledge, then it might not be possible for hearing professionals or non-native signers to code results similar to the coders in this study.

Response 20# This point is very relevant. The scoring has been thought in such a way that hearing signers, speech therapists, schoolteachers could score without help of native signers. As we said previously, coders were all hearing signers. Moreover we created a scoring booklet in which we explain the manner how we scored and which relevant linguistic facts the coder have to focus on.

Comment 21# Editorial feedback: Line 111, SLI was mentioned but it should be spelled out the first time used for the naïve reader. 

And, 132, “SLI children” should be changed to “children with SLI” (also, see line 112). 

Specific Language Impairment was spelled out (without its abbreviation) in the final section, Future Work. Please consider revising.

Response 21# All changes were done.

Comment 22# Claims made the sentence that starts with line 190 should be cited.

Response 22# we went back to line 190 the reference initially mentioned at line 195.

Comment 23#The reviewer did not review earlier submissions of the manuscript but read the authors’ responses to the past reviewers’ comments. It appears that the authors addressed most of the past reviewers’ concerns. The manuscript does appear to be more focused on test development and validity than phonology. The authors do not talk about validity at all although their results provide partial support on the validity of the LSF-SRT that they developed.

Response 23# We agree that our results partially support the validity of the LSF SRT as reflected by main effect of CA and AOA. However, we preferred refrain firm conclusion on validity issue before to have more data in further research.

We add the following sentence “Our results provide partial support of LSF SRT validity as reflected by different significant effect of CA and AOA. However, we refrain to formulate firm conclusions on the validity issue before to have at hand additional data in further research.” Line 683.

Answers to the Reviewer 2 request 

Reviewer #2: The manuscript has the potential to offer a valuable addition to research into LSF, and the use of a sentence repetition task as an assessment tool. It contributes to and confirms previous findings, whilst helping to bring LSF study in-line with studies in other sign languages. The notable number of native LSF participants from deaf families included in the study is also commendable.

However, I believe there are several issues that need to be addressed by the authors before it would be suitable for publication.

General comments:

Comment 24#I propose that the article be proofread and corrected by a native English speaker. Whilst the English is mostly clear and understandable, it is not currently at a suitable standard for publishing in PLOS ONE, with many errors and omitted words, so substantial editing is needed.

Response 24# We agree and the revised version was proofread and corrected by a native speaker of English. 

Comment 25# Care needs to be taken with consistency in using abbreviations. e.g. LSF, SL and SLI are frequently used in both long and short form. e.g. line 180-181 “SL development” is immediately followed by “sign language development”

Response 25# We decided to use long form for sign language and short from for SRT, SLI and all acronyms for sign languages of each countries (ASL, LSF, etc.). We checked for consistency along the whole manuscript.

Comment 26# Sections of the manuscript appear to be repetitious. Eg. the start of the discussion section, whilst intended as a recap for the reader, is very similar to the abstract.

Response 26# We agree that the first sentence of the discussion was similar with the sentence in the present study section. We reformulated this sentence in order to avoid repetition for readers. We replace by the following sentence in the revised sentence: 

Line 527 “This study aimed to assess LSF abilities in children to trace their developmental trajectory. For this purpose, different levels of analysis (phonological and morphosyntactic development, lexical knowledge) were investigated as a function of AOA and CA using an SRT”

Introduction

Comment 27# Line 111 is the first mention of ‘SLI’ so please correct to “specific language impairment (SLI)” for clarity.

Response 27# Thank you for this remark. We made the change accordingly.

Comment 28# Line 88 - Likewise, explain ‘ASL’ as ‘American Sign Language’ in the first instance for the reader, unless it can be assumed the readers will be familiar with the abbreviation.

Response 28# Thank you for this remark. We made the change accordingly

Methodology

Comment 29# Little information is given as to how the sentences for the SRT were created. Did the authors adapt sentences from other SRTs, or did they create new sentences? 

A pilot study is mentioned (line 313) from which 20 sentences were selected for their “naturalness”. I am unclear as to what is meant by this and think it requires some clarification. Given that the focus of the manuscript is the utilisation of an SRT, more information about the creation of the sentences would be useful. The description provided of the linguistic content of sentences, along with complexity level is good.

Response 29 # Thank you for this fruitful commentary which was also made by the other reviewer. Here, the answer proposed to the request information: “To create our 35 LSF sentences, we initially relied on the corpus of Rinaldi et al. (2018). We adapted the Italian version of the SRT thanks to deaf native signers in order to create a well-designed LSF culturally and linguistically pool of sentences. We added this information in the Stimuli section of the revised version: “These 20 sentences were selected from a pilot study in which 2 deaf adults assessed the initial pool of 35 LSF sentences based on their naturalness’. Line 329.

Concerning the naturalness, four criteria were taken into consideration: 1) grammaticality, 2) plausibility, 3) age-matched semantic content, 4) saliency of signs for display on screen. We added this sentence in the revised version” line 331

Comment 30# Participants - the inclusion of 10 deaf native signing adults is a useful addition to the study for comparison, however limited information is given about them other than their age range. As they are being used as a comparative group, please provide more detail on their ages (and mean age), and sex etc.

Response 30# We added the following sentences in the revised manuscript : “They are aged from 25 to 45 (29;9 years old, X Male) and grew up in different area from France, ensuring a real LSF expertise. Moreover, participants were match on the socio-economic status”

Comment 31# Procedure - line 337 states testing took place in the school library, suggesting that all of the participants attended the same school. If this is the case, it may be a relevant point to make for the study as it offers some control to the language environment/experience of all of the children whilst they are at school. It may be a strength to the study to make this clear, if it is the case.

Response 31 # Thank you for this suggestion. We added this information about native children in the Participant Section. We added the following sentence: “Note that all native children attended the same school, which ensured consistency with respect to the language environment/experience for all of them” Line 295.

Comment 32#If the children do all attend the same school, Table 1 on page 14 detailing children’s linguistic environment should clarify that this refers to their “home language environment”.

Response 32 # Unfortunately, all the children did not attend the same school. For this reason, we found more appropriate to take as language environment reference the home language environment especially for non-native signers in order to assess LSF exposure. 

Comment 33# Line 380-381 First states there are 6 coders, then states there are 2 coders - I assume the 2 independent coders were used for inter-rater reliability, but this is slightly unclear, please rephrase and add the reliability rating if available. Same request was made by the other reviewer concerning the intercoder reliability. 

Response 33# Thank you for this remark. It was a mistake for us. That was not six and we replace six by two.

we agree this is a relevant information to add. 

Analysis with Student t-test shows that they were no significant cross-coder differences for the following comparisons: 

- % of repeated signs (coder 1: 89.5% vs. coder 2: 90%), t(122) = 0.2, p > .05

- % of errors in repeated signs (coder 1: 41.8% vs. coder 2: 42.6%) t(122) = 0.24 p > .05

- Number of phonological errors (coder 1: 23.3% vs. coder 2: 25.6%), t(122) = 0.56, p > .0

- % errors of handshape (coder 1: 11.6% vs. coder 2: 12.2%), t(122) = 0.66, p > .05

- % errors of movement (coder 1: 11.1 % vs. coder 2: 12.1%), t(122) = 1.11, p > .05

- % errors of location (coder 1: 8.8%. vs. coder 2: 6.7%), t(122)=1.10, p > .05

Results

Comment 34# he asterisks indicating significance levels appear to be missing from the figures.

Response 34 # We have considered this suggestion. However, due to huge information in each figure, it will be too unreadable. We propose to add the name of each factor and interaction with the significant level.

Comment 35# Lines 431 and 432 contain commas instead of decimal points (2,84% instead of 2.84%). Proofreading needed.

Response 35 # Thank you for this remark. We made the change accordingly

Discussion

Comment 36# Lines 671-674 appear to state that there are no known developmental stages of language acquisition in deaf children and that the current study is a first attempt to address this issue. This statement is not true. I assume that the authors intend this statement to relate to LSF, but it is unclear to the reader so needs to be clarified or rewritten.

Response 36 # Thank you for this remark. We added in the revised version “”in LSF” to clarify this point. 

Comment 37# lines 671-674 are, in fact, in relation to LSF, I am not convinced that this study addresses the question of developmental stages of language acquisition in LSF, beyond the finding that native or early LSF acquisition results in higher accuracy on the task.

Response 37 # We agree that with respect to AOA, it cannot be viewed as a developmental stages study. Nevertheless, the study of AOA allow us t examine to which extent language development may be influence by AOA. However, as we varied the CA, this study constitutes an approach to discuss the developmental trajectory. 

Comment 38# Line 650 - the participants’ fluency was rated by coders on a scale of 0-5. Were the coders blind to which group the participants were in (native or late learners), and was any inter-rater reliability undertaken? Only the average score for early signers is provided in the text, please provide the rest of the data for this measurement.

Response 38 # Yes, the coders had no information about demographic data of learners. We added the following sentence “Coders did not know whether participants to rate were native or not.” The inter-rater reliability did not differ significantly. We add the average score for non-native signers in the revised manuscript. t(122)=0.20, p >.05

Mean C1 3,29

Mean C2 3,26

Comment 39# Lines 539-540 regarding regionalisms - were participants all tested in the same region, or from different areas of France? if participants were all from the same area (or school) would the authors predict differing regionalisms in their signing?

Response 39 # Thank you for this remark. No, the participants did not come from same region. Native children came from the same school in Toulouse (south of France), and non-native came from North area: Rouen and Paris 

For native group, we predict that the same regionalisms may be used.

---

## [Decision Letter · Decision Letter 1]

14 Jul 2020

Language experience in LSF development: Behavioral evidence from a sentence repetition task

PONE-D-20-04317R1

Dear Dr. Bogliotti,

We’re pleased to inform you that your manuscript has been judged scientifically suitable for publication and will be formally accepted for publication once it meets all outstanding technical requirements.

Kind regards,

Bencie Woll, PhD

Academic Editor

PLOS ONE

Additional Editor Comments (optional):

The authors have dealt with the reviewers’ comments and the paper now reads well and makes an important contributionn to studies of LSF acquisition. There still remain a few minor points made by the reviewers that should to be corrected in the final version.:

A few minor grammatical errors:

Line 22 - should read "10 adult signers were evaluated" instead of "was evaluated"

Line 42 - should read "on a case by case basis"

Line 56 - insert 'be' so the sentence reads "may be disturbed"

Line 65 - should read "no SRT study has..." instead of had.

Inconsistency in the use of ‘sign language’ vs SL.

Reviewer 2’s Comment 36. Please add a brief clarification.

The authors should briefly address Reviewer 1’s comment about the importance of the home spoken language.

The reporting of intercode comparisons should briefly report the correlation.

Reviewers' comments:

Reviewer's Responses to Questions

**Comments to the Author**

1. If the authors have adequately addressed your comments raised in a previous round of review and you feel that this manuscript is now acceptable for publication, you may indicate that here to bypass the “Comments to the Author” section, enter your conflict of interest statement in the “Confidential to Editor” section, and submit your "Accept" recommendation.

Reviewer #1: All comments have been addressed

Reviewer #2: (No Response)

2. Is the manuscript technically sound, and do the data support the conclusions?

Reviewer #1: Yes

Reviewer #2: Yes

3. Has the statistical analysis been performed appropriately and rigorously? 

Reviewer #1: Yes

Reviewer #2: Yes

4. Have the authors made all data underlying the findings in their manuscript fully available?

Reviewer #1: Yes

Reviewer #2: Yes

5. Is the manuscript presented in an intelligible fashion and written in standard English?

Reviewer #1: Yes

Reviewer #2: Yes

6. Review Comments to the Author

Reviewer #1: The authors developed a sentence repetition task in LSF and provided data on how native and non-native deaf signers performed on the test. They included age of LSF acquisition, chronological age, and error types in the analyses. The results shed light on the LSF fluency of deaf children of different language experiences and provide psychometric support for the use of the test in research and clinical settings.

I had the privilege to review the previous submission and this revision. I am impressed, overall, how much the authors took into consideration the reviewers comments in each manuscript revision. The present manuscript is much better than the previous one and I hardly have any new criticisms of the draft.

I think the sample could be more "clean" (and replicable) if the native signers had the same LSF experience at home and the non-native signers had the same French experience at home. I disagree with the authors' response that since the participants are deaf, the home spoken language is irrelevant. In the USA, deaf children from hearing English-speaking and Spanish-speaking families perform differently in school. While this does not mean that home language itself might have an effect but other factors that are associated with not having the dominant spoken language at home might have an effect on children's overall language development. I believe that if the two samples (native, non-native) could be more homogeneous within group by omitting those with different language experiences than the rest in the group. If such participant exclusion does not change the results of the study then this would improve the SRT replicability and cross-linguistic comparisons in future studies. The authors appear to disagree.

I would like to thank the authors for computing intercoder comparisons. I am more used to seeing the correlation being reported, rather than a t-test, to show the inter-rater reliability. Please provide the correlation as well in the final manuscript.

Reviewer #2: The authors have addressed all of the reviewers comments and cleared up the few previous areas of confusion. I would like to congratulate the authors on an interesting paper which will be a valuable addition to studies in LSF.

There remain only very minor editorial issues to be addressed from the previous review:

Comment 24 - the manuscript has been corrected by a native English speaker and the clarity of the manuscript has been much improved. There are only a few minor grammatical errors that remain:

Line 22 - should read "10 adult signers were evaluated" instead of "was evaluated"

Line 42 - should read "on a case by case basis"

Line 56 - insert 'be' so the sentence reads "may be disturbed"

Line 65 - should read "no SRT study has..." instead of had.

Comment 25 - the authors say that they have chosen to use the long-form of "sign language" (as opposed to the abbreviation SL) thought the manuscript. However, from page 4 onwards, a mixture of both forms are still used throughout.

Comment 36 - I acknowledge and appreciate that the authors have clarified that they are specifically referring to LSF in line 698. However, the previous sentence which begins the paragraph on line 695-697 still needs to be clear that they are referring to LSF and not language in general, which is how the sentence currently reads.

7. PLOS authors have the option to publish the peer review history of their article (what does this mean?). If published, this will include your full peer review and any attached files.

Reviewer #1: **Yes: **pH

Reviewer #2: No

---

## [Editor Report · Acceptance letter]

28 Jul 2020

PONE-D-20-04317R1 

Language experience in LSF development: Behavioral evidence from a sentence repetition task 

Dear Dr. Bogliotti:

I'm pleased to inform you that your manuscript has been deemed suitable for publication in PLOS ONE. Congratulations! Your manuscript is now with our production department. 

Kind regards, 

on behalf of

Professor Bencie Woll 

Academic Editor

PLOS ONE